# Ice-proximal sea-ice reconstruction in Powell Basin, Antarctica since the Last Interglacial

Wee Wei Khoo[1*], Juliane Müller[1,2,3], Oliver Esper[1], Wenshen Xiao[4], Christian Stepanek[1], Paul Gierz[1], Gerrit Lohmann[1,3,5], Walter Geibert[1], Jens Hefter[1] and Gesine Mollenhauer[1,2,3]

[1]Alfred Wegener Institute, Helmholtz Centre for Polar and Marine Research, Bremerhaven, Germany
[2]Department of Geoscience, University of Bremen, Bremen, Germany
[3]MARUM - Center for Marine Environmental Sciences, University of Bremen, Bremen, Germany
[4]State Key Laboratory of Marine Geology, Tongji University, Shanghai, China
[5]Department of Environmental Physics, University of Bremen, Bremen, Germany

**Correspondence:** Wee Wei Khoo (wee.wei.khoo@awi.de)

**ABSTRACT.** In Antarctica, the presence of sea ice not only plays a critical role in the climate system but also contributes to enhancing the stability of the floating ice shelves. Hence, investigating past ice-proximal sea-ice conditions, especially across glacial-interglacial cycles, can provide crucial information pertaining to sea-ice variability and deepen our understanding of ocean-ice-atmosphere dynamics and feedback. In this study, we apply a multiproxy approach, in combination with numerical climate modeling, to explore glacial-interglacial environmental variability. We analyze the novel sea ice biomarker $IPSO_{25}$ (a di-unsaturated highly branched isoprenoid (HBI)), open-water biomarkers (tri-unsaturated HBIs; z-/e-trienes), and the diatom assemblage and primary productivity indicators in a marine sediment core retrieved from Powell Basin, NW Weddell Sea. These biomarkers have been established as reliable proxies for reconstructing near-coastal sea-ice conditions in the Southern Ocean, where the typical use of sea ice-related diatoms can be impacted by silica dissolution. We present the first continuous sea-ice records, in close proximity to the Antarctic continental margin, since the penultimate deglaciation. Our data shed new light on the (seasonal) variability of sea ice in the basin, and reveal a highly dynamic glacial-interglacial sea-ice setting characterized by significant shifts from perennial ice cover to seasonal sea-ice cover and open marine environment over the last 145 kyrs. Our results also unveil a stronger deglacial amplitude and warming during the Last Interglacial (MIS 5e) compared to the current one (Holocene). A short-term sea ice readvancement also occurred towards the end of each deglaciation. Finally, despite similar findings between the proxy and model data, notable differences persist between both interglacials – emphasizing the necessity for different Antarctic ice-sheet configurations to be employed and more robust paleoclimate data to enhance climate model performance close to the Antarctic continental margin.

# 1 Introduction

Sea ice plays a vital role within Earth's climate system, exerting significant influence on air-sea interactions, ocean circulation and ecosystem dynamics. Its presence alters the surface albedo of the ocean through the reflectance of incoming solar radiation, thereby minimizing ocean warming (Ebert et al., 1995). Likewise, sea ice affects the atmosphere-ocean interaction by inhibiting the exchange of heat, gas and water vapor between both media (Dieckmann and Hellmer, 2010). During sea-ice formation, brine rejection occurs and leads to the production of high-saline shelf water. This dense high-saline shelf water then sinks towards the deeper ocean. Consequently, this process results in redistribution of salinity within the water column and has a profound impact on the stratification and ventilation of the ocean (Vaughan et al., 2013). For example, in a few locations in the Southern Ocean (SO), such as the Weddell Sea, the high-saline shelf water – depending on its route and mixing process – becomes the precursor of Antarctic Bottom Water (AABW), which is a major driver of the global thermohaline circulation and an important water mass that ventilates the deep ocean basins (Naveira Garabato et al., 2002; Rintoul, 2018; Seabrooke et al., 1971). Furthermore, when sea ice melts, the freshened surface water mixes with the upwelled deep water, contributing to the mode and intermediate waters in the Atlantic, Indian and Pacific sectors of the SO (Abernathey et al., 2016; Pellichero et al., 2018). Sea ice also serves as a crucial buttressing force at the ice front, effectively preventing or delaying the occurrence of potential calving events (Robel, 2017). This phenomenon was evident at locations such as the Mertz Glacier Tongue (Massom et al., 2015) and the Totten Ice Shelf (Greene et al., 2018) in East Antarctica. Furthermore, the presence of a sea-ice buffer in front of the ice terminus acts to diminish ocean swells as they propagate towards land. For instance, Massom et al. (2018) observed a substantial increase (orders of magnitude) in wave energy experienced at the fronts of the Larsen ice shelves and the Wilkins Ice Shelf when the sea-ice buffer was removed. In this regard, any changes to the sea-ice cover can potentially alter ice-ocean-atmosphere dynamics and ocean circulation patterns, making analyses of sea-ice variability over glacial-interglacial cycles, covering periods of less and more pronounced sea-ice cover, crucial.

Presently, numerous methods are used to reconstruct past sea-ice conditions, including biogenic proxies (e.g., biomarkers, diatoms, dinoflagellate cysts, foraminifera and ostracods) and sedimentological proxies (e.g., ice-rafted debris) in marine sediments, as well as chemical compounds archived in ice cores (e.g., methanesulfonic acid and sea-salt ($ssNa^+$); de Vernal et al., 2013 and references therein). Determination of methanesulfonic acid or $ssNa^+$ concentrations in Antarctic ice cores permits well-dated and temporally high-resolution regional sea-ice reconstructions but is often affected by other sea ice independent factors such as atmospheric transport (Abram et al., 2013). In particular, direct proxies, originating from sea-ice dwelling microorganisms, which are preserved in marine sediments are often preferred as they increase the reliability of sea-ice estimation (Leventer, 1998). Despite this, our understanding of past sea ice changes in the SO remains limited. The Cycles of Sea-Ice Dynamics in the Earth System working group (C-SIDE; Chadwick et al., 2019; Rhodes et al., 2019) consolidated a list of published Antarctic marine sea-ice records, as outlined in the review paper by Crosta et al. (2022). The compilation documents 20 studies on sea-ice variability during the Holocene (0-12 ka before present (BP)), 150 records detailing changes at the Last Glacial Maximum (LGM; ca.

21 ka BP or Marine Isotope Stage (MIS) 2), and a mere 14 sea-ice records dating back to around 130
ka BP. Notably, just two records extend beyond MIS 6 (ca. 191 ka BP; see also Fig. 3 in Crosta et al.,
2022). Their work underscores the pronounced dearth of (paleo) sea-ice reconstructions, particularly in
regions south of 60°S, notably in the Atlantic sector, and during the Last Interglacial (LIG) and beyond.
This scarcity of records, in particular proximal to the continental margin, is attributable to difficulties in
recovering marine sediment cores in the polar regions that at present are still subject to heavy year-
round ice cover, and a lack of continuous sedimentary records due to erosion and disturbance at the
sea floor during past glaciations. Moreover, limited preservation potential of silica frustules in SO
regions outside of the opal belt further hampers sea-ice reconstructions using diatom assemblages
(Ryves et al., 2009; Vernet et al., 2019). As such, important feedback mechanisms related to the sea
ice-ice shelf system during warmer-than-present periods and throughout climate transitions, remain
poorly understood. Ultimately, this lack of knowledge on how Antarctic ice sheets/shelves respond(ed)
to oceanic forcing may disadvantage climate models' ability to faithfully reproduce dynamics in the
ocean-sea ice-ice system, and limit our confidence in future projections of the Antarctic Ice Sheet's
contribution towards global sea level rise (Deconto and Pollard, 2016; Naughten et al., 2018). Despite
similar LIG winter sea-ice (WSI) retreats in marine records, inconsistency with regard to the position of
the sea-ice edge, in particular in the Atlantic sector, remains evident when the proposed spatial structure
of the $\delta^{18}$O-agreed WSI extent is compared to published marine records (Holloway et al., 2017).
Holloway et al. (2017) and Crosta et al. (2022) opined that this discrepancy may result from the marine
records (Bianchi and Gersonde, 2002; Chadwick et al., 2020; 2022) being located too far north to
adequately validate the $\delta^{18}$O-agreed WSI extent. Thus, they emphasized the need for additional marine
records closer to the continental margin to adequately constrain the spatial pattern of the LIG sea-ice
extent.

In recent years, the use of a novel sea-ice biomarker has been found increasingly applicable as a
suitable proxy for Antarctic sea-ice reconstructions (Belt et al., 2016; Smik et al., 2016). This sea-ice
biomarker, a di-unsaturated $C_{25}$ highly branched isoprenoid (HBI) alkene, introduced as an Antarctic
sea-ice proxy by Massé et al. (2011), was later termed Ice Proxy for the Southern Ocean with 25 carbon
atoms (IPSO$_{25}$), drawing parallel to the Arctic IP$_{25}$ (Belt et al., 2016). IPSO$_{25}$ is a lipid molecule produced
by the sympagic diatom *Berkeleya adeliensis*, which lives in the sea-ice matrix and is generally
abundant during late spring and early summer (Belt et al., 2016; Riaux-Gobin and Poulin, 2004), hence,
making the biomarker a good indicator for spring/summer sea ice. Furthermore, the biomarker is well-
preserved in the sediment and widely identified in areas near to the Antarctic continent (for more details,
see Belt, 2018; Belt et al., 2016). Nevertheless, there remains a risk of under-/overestimating the
presence of sea ice when relying solely on the IPSO$_{25}$ proxy. Thus, Vorrath et al. (2019) proposed
combining open-water phytoplankton markers like dinosterol or a HBI-triene with the IPSO$_{25}$ proxy, to
calculate the phytoplankton-IPSO$_{25}$ index (PIPSO$_{25}$). This enhances the quantitative application of the
IPSO$_{25}$ proxy. For example, in cases where the IPSO$_{25}$ concentration is minimal or absent, this may
imply either an open ocean condition (substantiated by a high phytoplankton signal) or the presence of
a perennial ice cover (evident by a low/absent phytoplankton signal). As such, the use of the PIPSO$_{25}$
proxy proves to be a more reliable approach to distinguish contrasting sea-ice settings (Belt and Müller,

2013; Lamping et al., 2020). To substantiate this application, Lamping et al. (2021) compared PIPSO$_{25}$-derived sea-ice estimates close to the Antarctic continental margin against satellite sea-ice observations and modeled sea-ice patterns, revealing strong correlation between the proxy, satellite and modeled data. Until now, the majority of HBI-based sea-ice reconstructions has focused on Holocene and deglacial/LGM time scales (e.g., Barbara et al., 2013; 2016; Denis et al., 2010; Etourneau et al., 2013; Lamping et al., 2020; Sadatzki et al., 2023; Vorrath et al., 2020, 2023) and one reconstruction dates back to the last ca. 60 ka BP (Collins et al., 2013). Yet, this tool has not been applied towards studying sea-ice variability in the Antarctic during warm climates beyond the current interglacial.

Here, we aim to investigate the glacial-interglacial environmental variability in the Powell Basin, NW Weddell Sea through a multiproxy approach, and provide the first continuous ice-proximal Antarctic sea-ice record covering the last ca. 145 kyrs. We present biomarker-based reconstructions of sea ice, subsurface ocean temperature, total organic carbon (TOC) and biogenic silica (bSiO$_2$) content, as well as diatom-based sea-ice concentration and summer sea surface temperature (SSST). This information is complemented by reconstructions of sea ice, primary productivity and SSST records from a neighboring core in the South Scotia Sea as well as numerically modeled sea ice, sea surface and subsurface temperatures to track latitudinal shifts in the environmental development in the Atlantic sector of the SO.

## 2 Study area

The Powell Basin (Fig. 1a) is a semi-isolated basin situated in the northwestern part of the Weddell Sea. It has an area of approximately 5x10$^4$ km$^2$ and an average water depth of 3.3 km (Coren et al., 1997; Viseras and Maldonado, 1999). The basin, enclosed by the Antarctic Peninsula to the west, the South Scotia Sea to the north, the South Orkney Microcontinent to the east, and the Weddell Sea to the South, is at present subject to the clockwise-circulating regime of the Weddell Gyre. As described in Orsi et al. (1993) and Vernet et al. (2019), the gyre involves four main water masses, namely Antarctic Surface Water, Warm Deep Water (WDW), Weddell Sea Deep Water (WSDW) and Weddell Sea Bottom Water (WSBW; Fig. 1b). The Antarctic Surface Water generally consists of shelf waters formed over the continental shelf, such as winter water, high salinity shelf water from brine rejection due to sea-ice formation, and ice-shelf water from glacial melt. The shelf waters travel along the Weddell Sea continental shelf via the Antarctic Coastal Current while denser shelf water cascades down and along the continental slope as the Antarctic Slope Current (Deacon, 1937; Fahrbach et al., 1992; Jacobs, 1991; Thompson et al., 2018). The WDW originates from the warm, saline and low-oxygen Antarctic Circumpolar Current that is advected and subsequently integrated into the gyre's circulation at its eastern front (Orsi et al., 1993; 1995). Along the southern boundary of the Weddell Gyre, the WDW upwells close to the Antarctic margin and mixes with the Antarctic Surface Water. The admixture cools and becomes denser, giving rise to the formation of WSDW and WSBW water masses at deeper water depths (Carmack and Foster, 1975; Dorschel, 2019; Huhn et al., 2008). In the Powell Basin, part of the WSDW flows out into the Scotia Sea through channels on the western slope of the basin (namely Philip,

Bruce and Discovery Passages; Morozov et al., 2020). The remaining WSDW and a part of WSBW
navigate around the southern and eastern South Orkney Plateau, progressing northward via the Orkney
Passage as AABW, while the residual WSBW recirculates within the Weddell Gyre (Fedotova and
Stepanova, 2021; Gordon et al., 2001; Orsi et al., 1999).

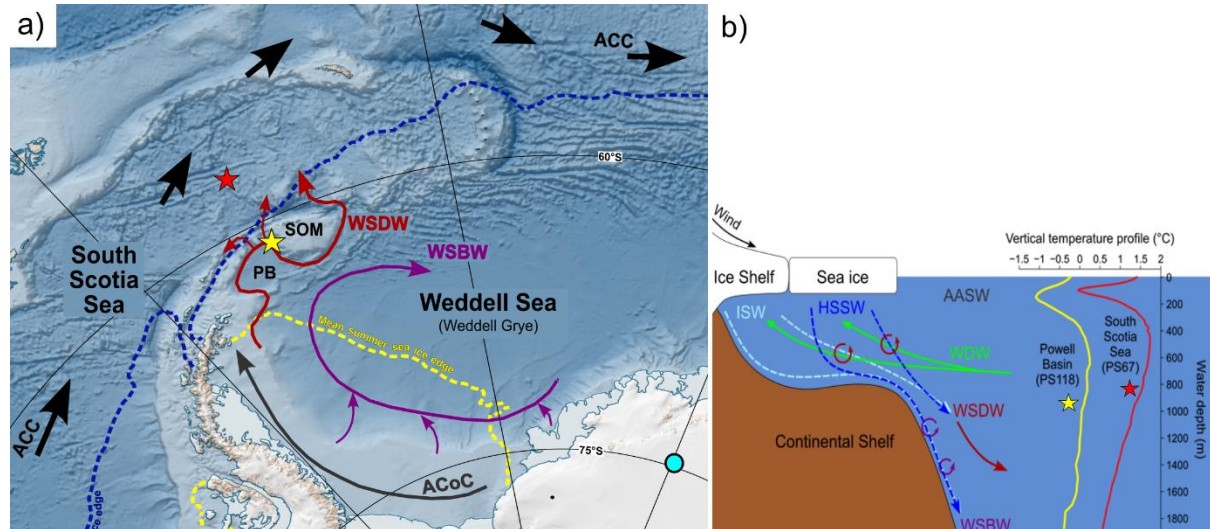

**Figure 1. a) Map of the study area showing the locations of marine sediment cores PS118_63-1 (yellow**
**star), PS67/219-1 (red star) and EDML ice core (light blue circle) discussed in this paper. Mean winter and**
**summer sea-ice extent (1981-2010; Fetterer et al., 2017) are illustrated by blue and yellow dotted lines,**
**respectively. Map was adapted from the Norwegian Polar Institute's Qantarctica package using QGIS 3.28**
**(Matsuoka et al., 2018). b) Diagram of the Weddell Gyre water masses with vertical spring/summer**
**temperature profiles collected near to our core sites in Powell Basin (-61.125ºS, -47.675ºW) and South**
**Scotia Sea (-57.125ºS, -42.375ºW; World Ocean Atlas 18; Locarnini et al., 2018). Pathways of ocean currents**
**(ACC: Antarctic Circumpolar Current – black; ACoC: Antarctic Coastal Current – grey) and water masses**
**(ISW: Ice Shelf Water – light blue; HSSW: High Saline Shelf Water – blue; WDW: Warm Deep Water – green;**
**WSDW: Weddell Sea Deep Water – red and WSBW: Weddell Sea Bottom Water – dark magenta) are**
**represented by the colored arrows. AASW: Antarctic Surface Water, PB: Powell Basin, SOM: South Orkney**
**Microcontinent.**
# 3   Materials and methods
## 3.1   Sediment core and age model
Gravity core PS118_63-1 was recovered from the Powell Basin during the RV *Polarstern* cruise
PS118 to the Weddell Sea in 2019 (Fig. 1a; Table 1; Dorschel, 2019). Physical properties, such as
magnetic susceptibility (MS) and wet-bulk density, were provided by Frank Niessen (shipboard data;
Dorschel, 2019). The age model of core PS118_63-1 is based on $^{14}$C radiocarbon dates, the
identification of the biostratigraphic marker *Rouxia leventerae*, as well as tuning against records from
the EDML ice core ($\delta^{18}$O and ssNa+) and nearby marine sediment core U1537 (MS, XRF-Fe and opal;
Weber et al., 2022). Refer also to Fig. 2 and Supplementary Table S2 for the tie points. Our age model
is further substantiated by age constraints of the uranium series disequilibrium, in particular the
constant-rate-of-supply model for $^{230}$Th-excess (Geibert et al., 2019). Further details on the
establishment of the age model and methods are provided in Supplement S1 and S2.
**Table 1. Locations and details of investigated and discussed cores.**

| Station | Latitude | Longitude | Water depth / Elevation (m) | Recovery (m) | Data source |
|---|---|---|---|---|---|
| **Marine sediment cores** | | | | | |
| PS118_63-1 | 61° 07.421'S | 47° 44.028'W | 2626.5 | 6.88 | this study |
| PS67/219-1 | 57° 13.22'S | 42° 28.02'W | 3619 | 20.71 | this study; Xiao et al, 2016a; Xiao et al, 2016b |
| **Ice core** | | | | | |
| EDML | 75°S | 0° | 2891 | | EPICA Community Members, 2006; Fischer et al, 2007 |


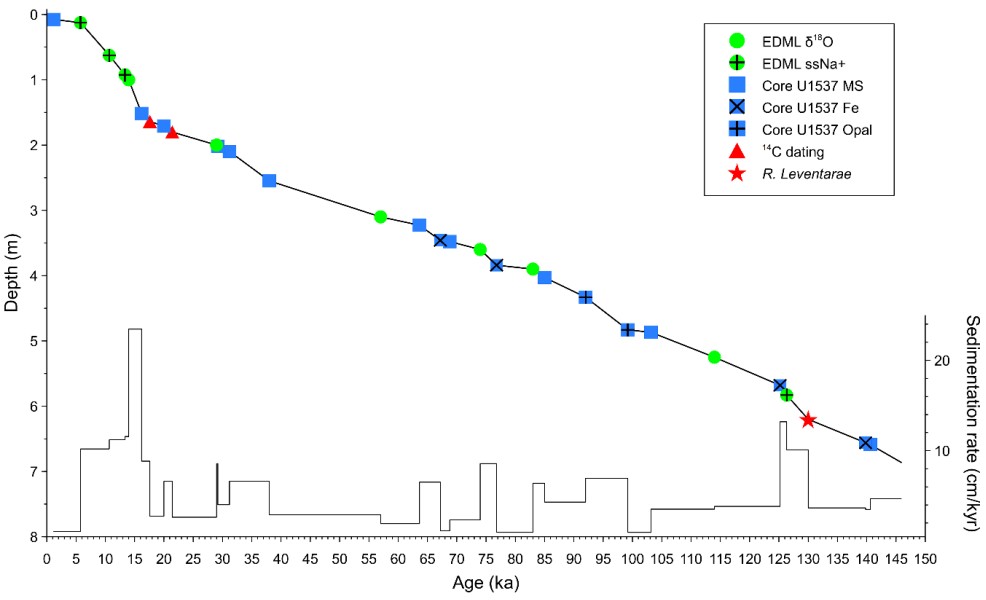

**Figure 2.Tie points used for the age-depth model of PS118_63-1 and sedimentation rates. EDML ice core**
**data is indicated by green circles, marine core U1537 data is marked by navy blue square, and available**
**AMS [14]C dates and the biostratigraphic marker (*R. leventerae*) from core PS118_63-1 are depicted by red**
**triangles ([14]C dates) and a red star (*R. leventerae*).**
**3.2    Bulk and organic geochemical analyses**
A total of 108 sediment samples, each with an approximate thickness of 1 cm, were collected from
core PS118_63-1. These samples were then freeze-dried and homogenized using an agate mortar and
pestle. All samples were stored in glass vials at -20 °C to prevent degradation. To analyze total organic
carbon (TOC), about 0.1 g of sediment was treated with 500 μL of hydrochloric acid to remove any
inorganic carbon, including carbonates. After the treatment, the TOC content was measured using a
carbon-sulfur analyzer (ELTRA CS800). Routine analyses of standard sediments were conducted
before and during each measurement yielding an error of ±0.02%. Biogenic opal was determined using
the automated continuous wet-chemical leaching method prescribed in Müller and Schneider (1993)
with an error of ±2 wt.%.For biomarker analyses, around 5-8 g of sediment were extracted and purified
in accordance with well-established protocols (Belt et al., 2012; Lamping et al., 2021). Prior to

extraction, internal standards, 7-hexylnonadecane (7-HND) and $C_{46}$-GDGT, were added for subsequent quantification of HBIs and glycerol dialkyl glycerol tetraether (GDGT) lipids. The biomarkers were extracted via ultrasonication (3 x 15 min) using DCM:MeOH (3 x 10 mL; 2:1 *v/v*) as solvent. Thereafter, the extracts were fractionated via open-column chromatography, with $SiO_2$ as the stationary phase, with the HBI-containing fractions eluted with 5 mL *n*-hexane and the GDGT fractions with 5 mL DCM:MeOH (1:1 *v/v*).

Compound analyses of HBIs were performed using an Agilent 7890B Gas Chromatograph (GC; fitted with a 30 m DB 1MS column; 0.25 mm diameter and 0.250 µm film thickness) coupled to an Agilent 5977B Mass Selective Detector (MSD; with 70 eV constant ionization potential, ion source temperature of 230°C). The GC oven temperature was first set to 60°C (3 min), then to 150°C (heating rate of 15°C/min), and finally to 320°C (heating rate of 10°C/min), at which it was held for 15 min for the analysis. Helium was used as the carrier gas. Specific compound identification was based on their retention times and mass spectra characteristics (Belt, 2018; Belt et al., 2000).

Quantification of each biomarker was based on setting the manually integrated GC-MS peak area relative to corresponding internal standards and instrumental-compound response factors. The concentrations were subsequently corrected to the extracted sediment weight. For HBI quantification, the molecular ions *m/z* 348 ($IPSO_{25}$) and m/z 346 (z-/e-trienes) were used in relation to its internal standard 7-HND (*m/z* 266). Finally, all biomarker mass concentrations were normalized to the TOC content of each sample. For calculating $PIPSO_{25}$, we adopted the equation as described in Vorrath et al. (2019):

$$PIPSO_{25} = IPSO_{25} / (IPSO_{25} + (\text{phytoplankton marker} \times c)), \tag{1}$$

where *c* is the ratio between the mean concentrations of $IPSO_{25}$ and phytoplankton marker and balances any significant offsets between both biomarker concentrations (Müller et al., 2011).

The GDGT fraction was dried ($N_2$) and redissolved in 120 µL hexane-isopropanol alcohol (99:1 *v/v*), followed by filtration through a polytetrafluoroethylene (PTFE) filter with 0.45 µm pore size membrane. GDGT measurement was performed using an Agilent 1200 series high-performance liquid chromatograph coupled to an Agilent 6120 atmospheric pressure chemical ionization mass spectrometer. Identification of isoprenoid GDGTs (isoGDGTs) and branched GDGTs (brGDGTs) was based on retention times and mass-to-charge ratios (isoGDGTs: *m/z* 1302, 1300, 1298, 1296 and 1292; brGDGTs: *m/z* 1050, 1036 and 1022). The late eluting hydroxylated-GDGTs (OH-GDGTs) with molecular ions *m/z* 1318, 1316 and 1314 were also determined during the scan of related isoGDGTs, namely *m/z* 1300, 1298 and 1296, respectively (Liu et al., 2012a; 2012b). The relative abundances were subsequently quantified relative to internal standard $C_{46}$ (*m/z* 744), instrumental response factors and the amount of sediment extracted. Mass content of all GDGTs were normalized to the TOC content of each sample.

The isoGDGT-based index, $TEX_{86}^{L}$ (Eq 2) was calculated following Kim et al. (2010) while the conversion to subsurface ocean temperature (OT; 0 - 200 m water depth; Eq 3) was conducted in accordance to Hagemann et al. (2023):

$$\text{TEX}_{86}^{L} = \text{Log}_{10} \frac{[\text{isoGDGT}-2]}{[\text{isoGDGT}-1]+[\text{isoGDGT}-2]+[\text{isoGDGT}-3]} \qquad (2)$$

OT (°C) = 14.38 x $\text{TEX}_{86}^{L}$ + 8.93; with a calibration error of ±0.6°C $\qquad$ (3)

The OH-GDGT-based index, RI-OH' (Eq 4) and the OT estimation (Eq 5) were determined following Lü et al. (2015). In their study, they determined that the RI-OH' is significantly correlated with temperature compared to other indices such as $\text{TEX}_{86}$ and RI-OH, producing a lower and less scattered residual sea surface temperature (SST) of ±6°C.

$$\text{RI-OH'} = \frac{[\text{OH}-\text{GDGT}-1]+2 \text{ x } [\text{OH}-\text{GDGT}-2]}{[\text{OH}-\text{GDGT}-0]+[\text{OH}-\text{GDGT}-1]+[\text{OH}-\text{GDGT}-2]} \qquad (4)$$

RI-OH' = 0.0382 x OT (°C) + 0.1 ($R^2$ = 0.75, n = 107, p <0.01) $\qquad$ (5)

The index of relative contribution of terrestrial organic matter against that of marine input (branched-isoprenoid tetraether, BIT; Eq 6) was calculated based on Hopmans et al. (2004):

$$\text{BIT} = \frac{[\text{brGDGT}-\text{I}]+[\text{brGDGT}-\text{II}]+[\text{brGDGT}-\text{III}]}{[\text{Crenarchaeol}]+[\text{brGDGT}-\text{I}]+[\text{brGDGT}-\text{II}]+[\text{brGDGT}-\text{III}]} \qquad (6)$$

Lastly, we utilize the ring index (RI; Eqs 7 - 9; Zhang et al., 2016) and methanogenic source indicator index (%GDGT-0; Eq 10; Inglis et al., 2015) to validate against possible non-thermal GDGT sources contribution:

$$\text{RI}_{\text{sample}} = 0\text{x}[\text{isoGDGT}-0] + 1\text{x}[\text{isoGDGT}-1] + 2\text{x}[\text{isoGDGT}-2] + \qquad (7)$$
$$3\text{x}[\text{isoGDGT}-3] + 4\text{x}[\text{crenarchaeol}] + 4\text{x}[\text{regio. crenarchaeol'}]$$

$$\text{RI}_{\text{calculated}} = -0.77 \text{ x } \text{TEX}_{86} + 3.32 \text{ x } (\text{TEX}_{86})^2 + 1.59 \qquad (8)$$

$$|\Delta\text{RI}| = \text{RI}_{\text{calculated}} - \text{RI}_{\text{sample}} \qquad (9)$$

$$\text{\%isoGDGT-0} = \frac{[\text{isoGDGT}-0]}{[\text{isoGDGT}-0]+[\text{Crenarchaeol}]} \text{ x } 100\% \qquad (10)$$

## 3.3 Diatom analyses

41 smear slides were prepared for a quantitative diatom assemblage analysis at respective depths of the core. Between 400-600 diatom valves, inclusive of those from *Chaetoceros* resting spores (*Chaetoceros* rs), were counted in each sample to ensure statistical significance of the results. Diatoms were identified to species or species group level and, if possible, to forma or variety level. The presence of sea ice is inferred from the percentage of sea-ice indicating diatoms. A combined relative abundance of *Fragilariopsis curta* and *Fragilariopsis cylindus* (hereon referred to as *F. curta* gp) of >3% is used as a qualitative threshold to represent presence of WSI, while values between 1 and 3% estimates the

edge of maximum winter sea ice (Gersonde et al., 2003; 2005). Likewise, *Fragilariopsis obliquecostata*
is used to indicate summer sea ice (Gersonde and Zielinski, 2000).
We reconstructed WSI concentration (WSIC) by applying a marine diatom transfer function
developed by Esper and Gersonde (2014b; TF MAT-D274/28/4an). This transfer function consists of
274 reference samples from surface sediments in the Atlantic, Pacific and western Indian sectors of the
SO, with 28 diatom taxa and taxa groups, and an average of 4 analogs (Esper and Gersonde, 2014b).
The WSI estimates refer to September sea-ice concentration averaged over a period between 1981
and 2010 at each surface sediment site (National Oceanic and Atmospheric Adminstration, NOAA;
Reynolds et al., 2002; 2007). The reference dataset fits our approach as it uses a 1° by 1° grid, providing
a higher resolution than previously used, and giving a root mean square error of prediction of 5.52%
(Esper and Gersonde, 2014b).
The SSST was estimated using TF IKM-D336/29/3q (standard error of ±0.86ºC), comprising 336
reference samples from surface sediments in the Atlantic, Pacific and western Indian sectors of the SO,
with 29 diatom taxa and taxa groups, and a 3-factor model calculated with quadratic regression (Esper
and Gersonde, 2014a). The SSST estimates refer to summer (January-March) temperatures at 10 m
water depth averaged over a time period from ≤1900 to 1991 (Hydrographic Atlas of the Southern
Ocean; Olbers et al., 1992). The Hydrographic Atlas of the Southern Ocean was used because it
represents an oceanographic reference dataset least influenced by the recent warming in the SO (Esper
and Gersonde, 2014a).
**3.4    Comparison with other proxy records**
The EDML ice core and the marine sediment core PS67/219-1 are used in this study for regional
comparison due to proximity of both cores to our core site (Fig. 1a; see also Table 1 for details). Water
isotope (δ$^{18}$O) and ssNa$^+$ records of the EDML ice core were investigated by EPICA Community
Members (2006) and Fischer et al. (2007), respectively. Marine sediment core PS67/219-1, retrieved
from the South Scotia Sea, is located south of the Polar Front and just north of the modern-day winter
sea-ice extent. This core offers data on sea ice, SSST and biogenic opal, which extend at least to the
LIG period, making it suitable for comparison with core site PS118_63-1. The chronology and biogenic
opal data of core PS67/219-1 was described and published in Xiao et al. (2016b), while investigations
on sea-ice reconstruction and SSST for the last 30 ka BP are presented in Xiao et al. (2016a). We
further extend the WSIC and SSST records, back to 150 ka BP, using the transfer functions TF MAT-
D274/28/4an and TF IKM-D336/29/3q, respectively (Esper and Gersonde, 2014b; 2014b).
**3.5    Comparison with simulations from climate model(s)**
Here, we also analyze model-simulated sea ice, SST and OT estimates for further comparison and
evaluation against our proxy results. In this respect, the strength of our modeling approach is twofold.
First, the model shall provide reasonable coverage of our intended studied time slices, mainly 6, 21,
125, 128 and 140 ka BP. Second, the model's sensitivity to various climate forcings and boundary
conditions across the Quaternary and the entire Cenozoic era must be known. To this end, the
Community Earth System Models (COSMOS; Jungclaus et al., 2006) is chosen over other climate
models due to its proven track record. For example, the simulation ensemble that has been produced
over the years with COSMOS is extensive and not available from international modeling initiatives like
the Paleoclimate Modeling Intercomparison Project (PMIP; e.g., Braconnot et al., 2012). Likewise, the
model has reproduced various aspects of reconstructed paleoclimate data (see Supplement S3.1 for a
list of paleo-studies using the COSMOS model), is shown to be sensitive to paleogeography and climate
forcing, and is being characterized by a large Climate and Earth System Sensitivity (Haywood et al.,
2013; Stepanek and Lohmann, 2012). Additionally, COSMOS has been proven useful for the study of
both warmer (Pfeiffer and Lohmann, 2016) and colder (Zhang et al., 2013; 2017) climates than today
and supported research in sometimes very interdisciplinary frameworks (e.g., Guagnin et al., 2016;
Klein et al., 2023). For some of the periods relevant here – Holocene, Last Glacial Maximum, LIG –
standalone applications of the model are documented (e.g., Pfeiffer and Lohmann, 2016; Wei and
Lohmann, 2012; Zhang et al., 2013). More importantly, results from COSMOS have been extensively
compared to other models, particularly within the framework of the PMIP, with a focus on the
Holocene (Dallmeyer et al., 2013; 2015; Varma et al., 2012) and the Last Interglacial (Bakker et al.,
2014; Jennings et al., 2015; Lunt et al., 2013). A relevant inference from comparing PMIP3-class models
is that, from the viewpoint of model performance in the SO, COSMOS has shown to be among the
models with a comparably minor warm bias in SST (see Fig. 4e and f in Lunt et al., 2013). This makes
COSMOS particularly suitable for the studies of ocean temperatures and sea ice around the Weddell
Sea. We refer to additional discussion on the rationale for choosing COSMOS over the PMIP models
in our study in the Supplement S3.3. Additionally, we also provide an in-depth comparison and
evaluation of the simulated results from PMIP3 and PMIP4 ensemble models, within the context of our
study, and agreement between COSMOS and PMIP ensemble models in the Supplement S3.4.
*3.5.1    Community Earth System Models*
In our study, the model data is derived from climate simulations performed with COSMOS. The
model's atmospheric module is the fifth generation of the European Centre for Medium-Range Weather
Forecasts' Model (ECHAM5), a model of the general circulation of the atmosphere, with a spectral
dynamical core, developed at the Max Planck Institute for Meteorology in Hamburg up to the sixth
generation (Stevens et al., 2013). In our model setup, the ECHAM5 is employed at a truncation of T31,
corresponding to a spatial resolution of approximately 3.75°x3.75°, or 400 km at the equator. The
atmospheric column is discretized at a resolution of 19 vertical hybrid sigma-pressure levels. The
ECHAM5 also encompasses a land surface component (JSBACH) that represents multiple land cover
classification types (Loveland et al., 2000; Raddatz et al., 2007). We employ JSBACH's capability to
reflect vegetation dynamics (Brovkin et al., 2009) in the course of climate simulations. In our setup, we
consider eight different plant functional types (see Table 1 in Stepanek and Lohmann, 2012) that the
model adapts in response to changes in the simulated climate, thereby reflecting important feedback
processes between vegetation and climate in our simulations (Stepanek et al., 2020). The ocean
module is the Max Planck Institute Ocean Model (MPIOM; Marsland et al., 2003), employed at 40
unevenly spaced pressure levels with a bipolar curvilinear GR30 grid that has a formal resolution of
1.8°x3.0°. This enables the horizontal resolution to reach grid cell dimensions that are as small as 29
km at high latitudes. Sea ice computation is based on dynamic-thermodynamic processes with viscous-
plastic rheology and follows the formulation by Hibler (1979). Various parameterizations improve the
representation of small-scale ocean dynamics in the simulations. For additional information about the
parameterizations utilized in our model setup, and for the steps taken to create geographic setups to
apply the model in paleoclimatological research, see, for example, Stepanek et al. (2020) and
references therein.
*3.5.2   COSMOS simulation settings*
The simulation ensemble consists of a pre-industrial reference state (simulation *piControl*, 1850
CE; Wei and Lohmann, 2012), a mid-Holocene climate (simulation *mh6k*, 6 ka BP; Wei and Lohmann,
2012), an LGM state (simulation *lgm21k*, 21 ka BP; Zhang et al., 2013), two time-slices of the LIG,
where one refers to conditions at 125 ka BP (simulation *lig125k*) and one to conditions at 128 ka
(simulation *lig128k*), and a Penultimate Glacial Maximum (PGM) climate (simulation *pgm140k*). In order
to filter out short-term climate variability on interannual and multidecadal time scales, and to derive
average climatic conditions that are representative of the respective Quaternary time-slice, we average
the modeled climate state over a period of 100 model years. For interglacial climates we employ a
modern geography. The boundary conditions for the Last and Penultimate Glacial Maximum have been
set up for a study by Zhang et al. (2013) based on the PMIP3 modeling protocol. Details of the ice-
sheet reconstruction, that is a blend of ICE-6G v2.0 (Argus and Peltier, 2010), ANU (Lambeck et al.,
2010) and GLAC-1a (Tarasov et al., 2012), are described by Abe-Ouchi et al. (2015). For further details
on the climate states and simulation configurations, we refer to the supplement (S3.2 and
Supplementary Table S3, respectively). For analysis, the climate model output is interpolated from the
native grid of the ocean model to a regular resolution of 0.25°x0.25°. High resolution is chosen in order
to preserve the geographic features of the ocean model. Additionally, we also derived climate model
data specifically tailored to the two marine core sites discussed in this paper, achieving this through
interpolating relevant climate fields to the geographic coordinates of each core using a nearest-neighbor
interpolation algorithm. Any reference to the modeled sea-ice edges in this publication specifies the
isoline of 15% of sea-ice cover.
# 4   Results
**4.1   HBIs**
The concentration of the sea-ice biomarker (IPSO$_{25}$; Fig. 3a) in core PS118_63-1 varies
significantly between 0 and 2.41 µg/g OC. Peak concentration is found at ca. 112 ka BP, while very low
concentrations are noted throughout MIS 2-4, 5d, 5e and 6. Moderate to low concentrations are
observed during MIS 1 and through both terminations. The concentration of the ice marginal-open water
phytoplankton biomarkers varies between 0 - 3.03 µg/g OC (z-triene) and 0 - 0.76 µg/g OC (e-triene;
Fig. 3b). Higher concentrations are observed at MIS 1 and 5e, while lower concentrations are noted
throughout MIS 2-4, 5d and 6. In our investigation, we utilized both z- and e-trienes, respectively, to
calculate the semi-quantitative spring/summer sea-ice indices (P$_{z/e}$IPSO$_{25}$). This combined use of

biomarkers, indicative of ice marginal-open water conditions and IPSO$_{25}$, helps to circumvent ambiguous interpretations especially when dealing with scenarios of permanent sea ice and open ocean conditions. Our P$_z$IPSO$_{25}$ index ranges between 0.09 and 1, while the P$_e$IPSO$_{25}$ index varies from 0.06 to 1 (Fig. 3c). Instances, where both values of IPSO$_{25}$ and z-/e-triene are zero, the P$_{z/e}$IPSO$_{25}$ index is assigned a value of 1, indicating permanent ice cover. Both index profiles presented a similar trend (r = 0.98), with higher values (>0.8) throughout MIS 2-4, 5d and 6, while reduced values are noted for MIS 1 and 5e. Notably, the lowest P$_{z/e}$IPSO$_{25}$ values (<0.2) are observed during MIS 5e, specifically between 119 and 128 ka BP, signifying a distinct decline in sea ice and more open ocean conditions during this time interval. Comparable low P$_{z/e}$IPSO$_{25}$ values are also observed around 4 and 12 ka BP.

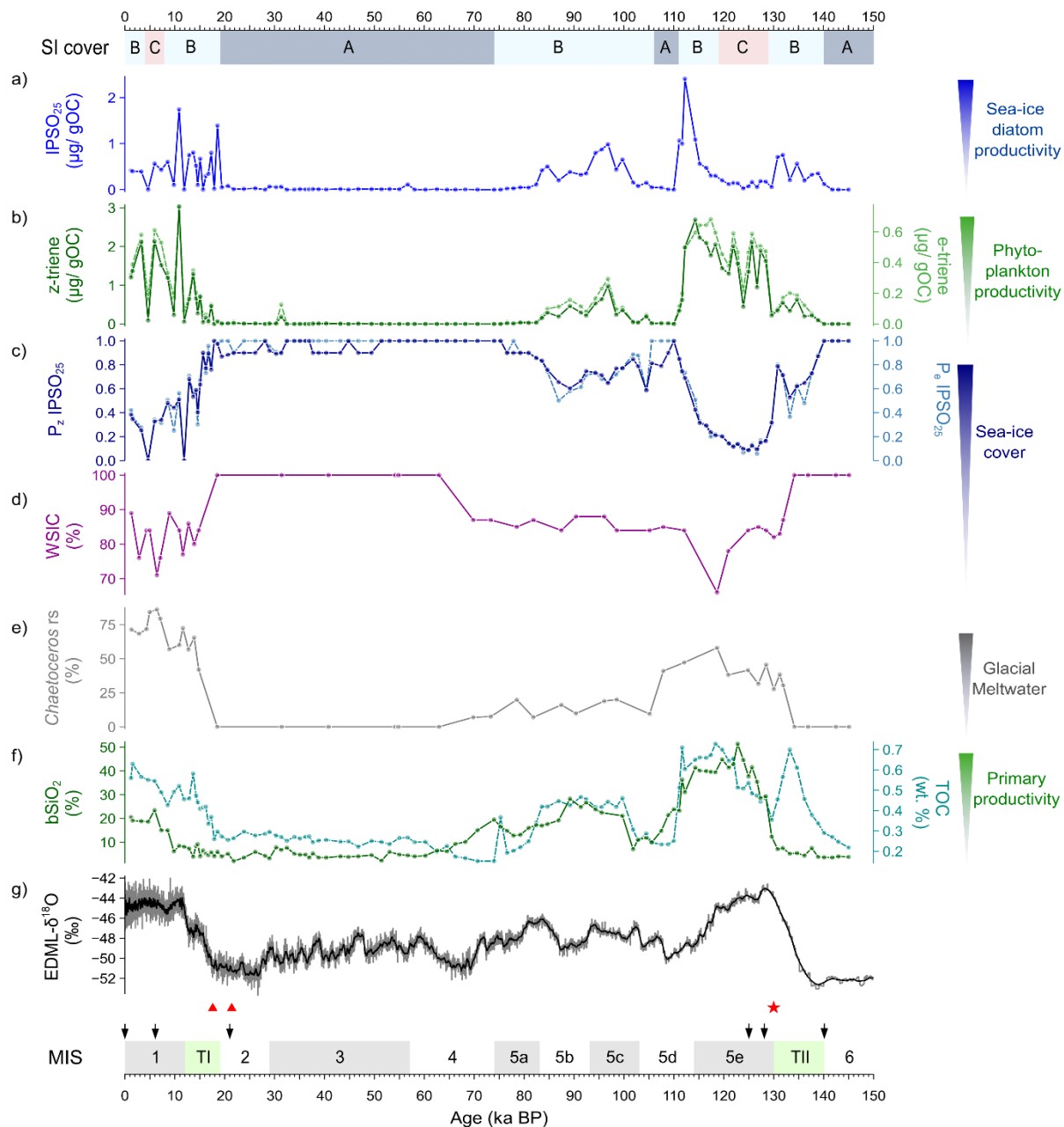

**Figure 3. Multiproxy analyses of sea-ice conditions in Powell Basin, reconstructed from marine sediment core PS118_63-1. Sea-ice (SI) cover scenarios: A - permanent sea-ice cover (dark blue), B - dynamic sea-ice cover (light blue) and C - minimal sea-ice cover (light red). From top to bottom: a) HBI-based sea ice biomarker (IPSO$_{25}$), b) HBI-based phytoplankton biomarkers (z-/e-trienes), c) Phytoplankton-IPSO$_{25}$ index**

**(PIPSO$_{25}$), d) Diatom-based winter sea-ice concentration (WSIC), e) Glacial meltwater indicator**
**(*Chaetoceros* resting spores) and f) Biogenic opal (bSiO$_2$), and total organic carbon (TOC). Atmospheric**
**temperature is implied by g) the $\delta^{18}O$ record from the EDML ice core. AMS $^{14}C$ dates are marked with red**
**triangles, the biostratigraphic marker (*R. leventerae*) is indicated by the red star. The black arrows**
**delineated the time-slices for the model simulations in this study. MIS stages are depicted in alternating**
**grey (odd) and white (even) shades, while the terminations TI and TII are shown in green.**

## 4.2 GDGTs

Downcore OT estimates using the RI-OH' index cover a temperature range between -2.5 and
1.0°C (Fig. 4g) while TEX$_{86}^L$-derived OT fluctuates between -2.6 and 1.0°C (Supplementary Fig. S5a).
These GDGT-based OTs likely reflect (mean) annual ocean temperature between the water depths of
0 and 200 m (Hagemann et al., 2023; Kim et al., 2012; Liu et al., 2020), and this seems to be
corroborated by the modern-day vertical ocean temperature profile nearby core site PS118_63-1 (Fig.
1b). Certainly, these minimum temperatures of less than -1.9°C – freezing temperature of seawater –
need to be considered with caution due to factors influencing the ocean temperature calibration, for
example, bias from terrestrial input, water depth, use of satellite-assigned ocean temperature below the
freezing point of seawater and inadequate samples from polar areas (Fietz et al., 2020; Xiao et al.,
2023). Nevertheless, both OT proxies consistently indicate a cold-water subsurface regime (0 – 200 m;
<1°C) with a 0-2°C temperature fluctuation, and no significant glacial/interglacial variability over the last
145 kyrs. We further note that the RI-OH'-based OTs fluctuate within the error range of the temperature
calibration based on a global surface sediment dataset (Lü et al., 2015) and call for attention when
interpreting OT variability. Calculation of terrestrial originated-GDGT (i.e. BIT) and isoGDGT-related
indices (i.e. %isoGDGT-0 and ΔRI; Supplementary Fig. S5b-e) reveals the presence of potential non-
thermal influences on the TEX$_{86}^L$ index, which may lead to bias in the temperature reconstruction (see
also S4 in the Supplement). In light of the non-thermal influences on TEX$_{86}^L$, we have decided not to
further discuss on the TEX$_{86}^L$-derived OT in this paper. Concerning the RI-OH' approach, the presence
of OH-GDGT has, thus far, only been observed within the cultivated marine thaumarchaeal group I.1a
(Pitcher et al., 2011; Liu et al., 2012b; Elling et al., 2014; 2015). Its absence in the terrestrial
thaumarchaeal group I.1b (Sinninghe Damsté et al., 2012) suggests a predominantly planktic origin (Lü
et al., 2015). While both isoGDGTs and OH-GDGTs are derived from the phylum *Thaumarchaeota*,
variances in their ring composition indicate that the OH-GDGTs may be biosynthesized from different
source organisms or differing conditions (Liu et al., 2012b). Additionally, previous studies compared the
relationship between various GDGT-based indices (i.e. RI-OH, RI-OH', TEX$_{86}$ and TEX$_{86}^L$) and
temperature, and determined that the RI-OH'-temperature relationship shows the most significant
correlation in cold-water (<15°C) regions, making the RI-OH' a robust temperature proxy for the
(sub)polar regions (Lü et al., 2015; Lamping et al., 2021; Park et al., 2019; Fietz et al., 2020). Therefore,
we suggest that the RI-OH' index holds promise as a potential OT proxy for our study site. However,
further work on the distribution of OH-GDGT and calibration studies are still essential to enhance the
applicability of RI-OH' as a (paleo)temperature proxy.

## 4.3 Diatoms

The diatom-based data for cores PS118_63-1 and PS67/219-1 are presented in Fig. 4c and d. For
core PS118_63-1 from the Powell Basin, the relative abundance of sea ice-related diatoms ranges
between 2 and 39% for *F. curta* gp, and from 0 to 6% for *F. obliquecostata*. The relative abundance of
diatoms between ca. 15 and 70 ka BP, and before 131 ka BP, is rare/absent (Fig. 4c). Such cases
generally indicate the presence of permanent sea ice over the core site (Zielinski and Gersonde, 1997).
We, therefore, assign the diatoms' relative abundance as 0, and WSIC as 100%, to above-mentioned
time intervals (i.e., MIS 2 - 4 and 6). The abundance of *F. curta* gp is noted to be above the 3% threshold
(indicative of presence of WSI) throughout the remaining time periods – except at 6 ka BP, where the
lowest abundance (2%) is observed. A relative abundance of *F. obliquecostata* around the 3% threshold
indicates a dynamic summer sea-ice edge over the area during MIS 1 and 5. The WSIC across the rest
of the time frame, namely MIS 1 and 5, is generally high (>75%) with a couple of lower WSIC observed
at ca. 6 ka BP (71%) and at 119 ka BP (66%). The abundance of *Chaetoceros* resting spores
(*Chaetoceros* rs) varies between 0 and 86%, with higher values noted during MIS 1 and 5e (Fig. 3e).
Such increases in the abundance of the *Chaetoceros* rs imply the presence of glacial meltwater at the
core location (Crosta et al., 1997). The diatom-derived SSST – typically indicating summer ocean
temperature between the water depth of 0 and 10 m – covers a temperature range between -0.8 and
0.4°C (Fig. 4h), and describes a cold-water region during MIS 1 and 5, similar to the RI-OH'-derived OT
(Fig. 4g).
To the north in the South Scotia Sea, core PS67/219-1 documents an overall lower percentage of
sea ice-related diatoms (Fig. 4d). Similar to core PS118_63-1, the relative abundance of *F. curta* gp
(0.5-20%) is noted to be mostly above the 3% threshold, indicating presence of WSI over the region,
with higher abundance observed for MIS 2 and 3, and lowest abundance (<1%) observed during MIS
5e. However, the relative abundance of *F. obliquecostata* for core PS67/219-1 remains below the 3%
threshold, between 0 and 3%, suggesting a lack of summer sea ice over the core site. The percentage
of WSIC in the South Scotia Sea is also lower than that of Powell Basin, with a record of 37-82%. The
diatom-based SSST documents a SSST range of -0.7 to 2°C, with colder SSST registered during MIS
2 and 3, and warmer SSST noted during MIS 1 and 5e (Fig. 4i).
**4.4  TOC and Biogenic opal**
In this study, both TOC and biogenic opal (Fig. 3f) are interpreted to reflect primary productivity (r
= 0.65). The TOC content varies between 0.2 and 0.7% while biogenic opal ranges from 2 to 51%.
Highest productivity is observed during MIS 1 and 5e, indicative of favorably warmer conditions that
promote primary productivity blooms at the core location. A rather moderate productivity level is
observed between MIS 5a to c, while lowest values are noted for MIS 2-4, 5d and 6. Both profiles also
exhibit some differences. For example, peak biogenic opal occurs around 124 ka BP whilst peak TOC
is recorded at 119 ka BP. We also observe a more pronounced increase in the TOC content during the
terminations than in the biogenic opal content. This is likely due to greater input from non-siliceous
organisms, such as archaeal, bacterial and terrestrial input (see Supplementary Fig. S4).

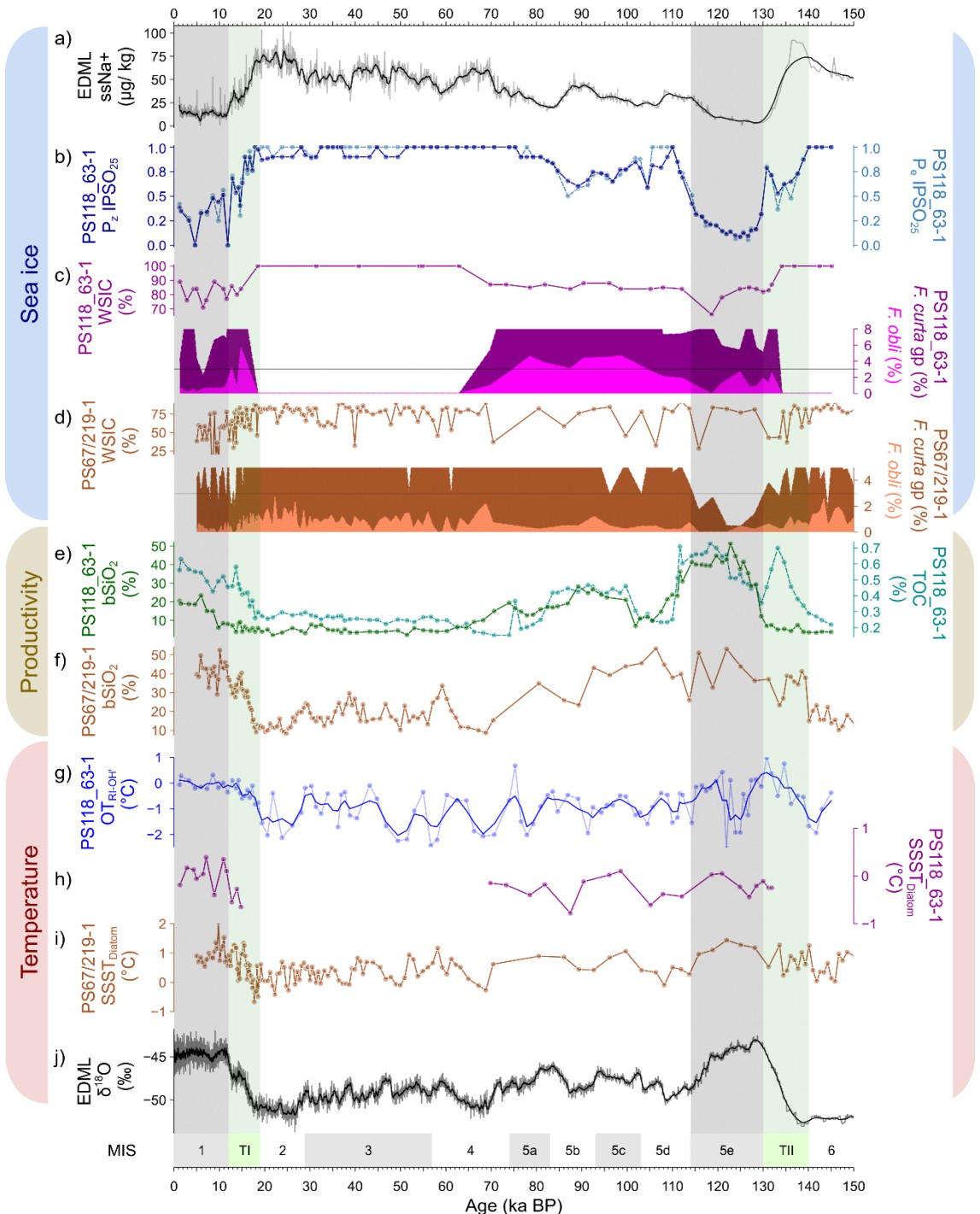


**Figure 4. Regional sea ice, productivity and temperature variability in the South Atlantic sector of the**
**Southern Ocean as inferred from EDML ice core, Powell Basin (PS118_63-1) and South Scotia Sea**
**(PS67/219-1). For sea ice: a) sea-ice estimation (ssNa+; black) from EDML ice core, b) HBI-based sea ice**
**indicator ($P_zIPSO_{25}$ – dark blue; $P_eIPSO_{25}$ – dotted light blue), c) diatom-based winter sea-ice concentration**
**(WSIC – dark magenta), *F. curta* group (*F. curta* gp – dark magenta), *F. obliquecostata* (*F. obli* – light**
**magenta) from PS118_63-1, and d) diatom-based WSIC (brown), *F. curta* group (*F. curta* gp – brown), *F.***
**obliquecostata (*F. obli* – light brown) from PS67/219-1. For productivity: e) biogenic opal ($bSiO_2$ – dark**
**green) and total organic carbon (TOC – dotted light green) from PS118_63-1 and f) $bSiO_2$ (brown) from**
**PS67/219-1. For temperature: g) RI-OH'-derived subsurface ocean temperature with three-point smoothing**
**($OT_{RI-OH'}$ – navy blue) and h) summer sea surface temperature ($SSST_{Diatom}$ – dark magenta) from PS118_63-**
**1, i) $SSST_{Diatom}$ (brown) from PS67/219-1 and j) EDML water stable isotope record ($\delta^{18}O$ – black). The 3%**
**threshold for diatom species relative abundance is indicated by a black horizontal line. MIS stages are**
**depicted in alternating grey (odd) and white (even) shades, while the terminations TI and TII are shown in**
**green. For the full *F. curta gp* abundance data, refer to the relevant datasets in Pangaea (refer to Data**
**availability).**

## 4.5  Sea-ice conditions – a multiproxy approach

Using a multiproxy approach, our analysis of the data from core PS118_63-1 provides a continuous glacial-interglacial sea-ice history in the Powell Basin since the PGM. We distinguish three different sea-ice scenarios spanning the last 145 kyrs (Fig. 3).

*A) Perennial sea-ice cover.* This scenario is characterized by remarkably low (sea ice) diatom abundances, minimum $IPSO_{25}$ and HBI-triene concentrations, as well as minimum $bSiO_2$ and TOC contents. We deduce the presence of maximum WSIC and spring/summer sea ice ($PIPSO_{25}$) cover. These results indicate a glacial setting, with our core site situated under a perennial sea ice or ice-shelf cover suppressing primary production in the water column. Such a scenario persisted throughout the glacial periods MIS 2-4, MIS 6, and during MIS stadial 5d.

*B) Dynamic sea-ice cover.* This scenario is described by fluctuations in each of the proxy profiles, in particular WSIC, $PIPSO_{25}$, HBI-trienes, $bSiO_2$ and TOC contents. These records reflect the dynamic nature of sea-ice conditions over our core site, with varied primary production at different time intervals. This scenario is prevalent during periods of climate transition, such as terminations I and II, and during MIS 1 and 5a-c.

*C) Minimal (winter-only) sea-ice cover.* This scenario is denoted by a considerably reduced sea-ice diatom ($IPSO_{25}$) production, WSIC and $PIPSO_{25}$, coupled with high phytoplankton productivity (HBI-trienes), $bSiO_2$ and TOC contents. These findings suggest that our core site experienced ice-free or winter-only ice conditions, permitting enhanced primary production in the water column. This scenario occurs in short time intervals within the MIS 1 and MIS 5e.

## 4.6  Inferences from COSMOS simulations

Covering the Atlantic sector of the SO, our model-simulated sea ice, SST and OT (at 220 m water depth) glacial-interglacial time-slices cover the PGM at 140 ka BP, LIG at 128 (sea ice only) and 125 ka BP, LGM at 21 ka BP, Holocene at 6 ka BP and pre-industrial (Fig. 5 - 7). In Fig. 5, the left column (Fig. 5a) shows the simulated sea-ice cover/extent for the spring/summer seasons (NDJFMA, this averaging period considers the time lag in sea-ice extent vs. spring/summer temperature evolution) while the right column (Fig. 5b) illustrates the simulated sea-ice cover/extent for the winter (ASO) season. In general, a greater sea-ice cover is observed during winter than spring/summer for each time-slice. During the glacial periods, the model highlights a northward expansion of the sea-ice extent beyond both marine core sites (PGM: Fig. 5.1; LGM: Fig. 5.4). At the more southern site (Powell Basin; core PS118_63-1), the modeled glacial sea-ice cover varies between ~93 to 94% (winter) and ~79 to 82% (spring/summer), while at the more northern site (South Scotia Sea; core PS67/219-1), sea-ice cover varies around ~91% (winter) and ~26 to 34% (spring/summer). In contrast, during the interglacials, fluctuations in sea-ice extent are more pronounced between seasons. WSI extent is observed to be located north of both core sites (Fig. 5.2b, 5.3b, 5.5b and 5.6b), with the WSI cover ranging between ~86 and 89% at core site PS118_63-1, and ~52 to 69% at core site PS67/219-1. During spring/summer, the sea-ice extent retreats to a latitude between both sites (Fig. 5.2a, 5.3a, 5.5a and 5.6a), with the spring/summer sea-ice cover varying from ~31 to 35% at core site PS118_63-1 and between ~0 and 4% at core site PS67/219-1.

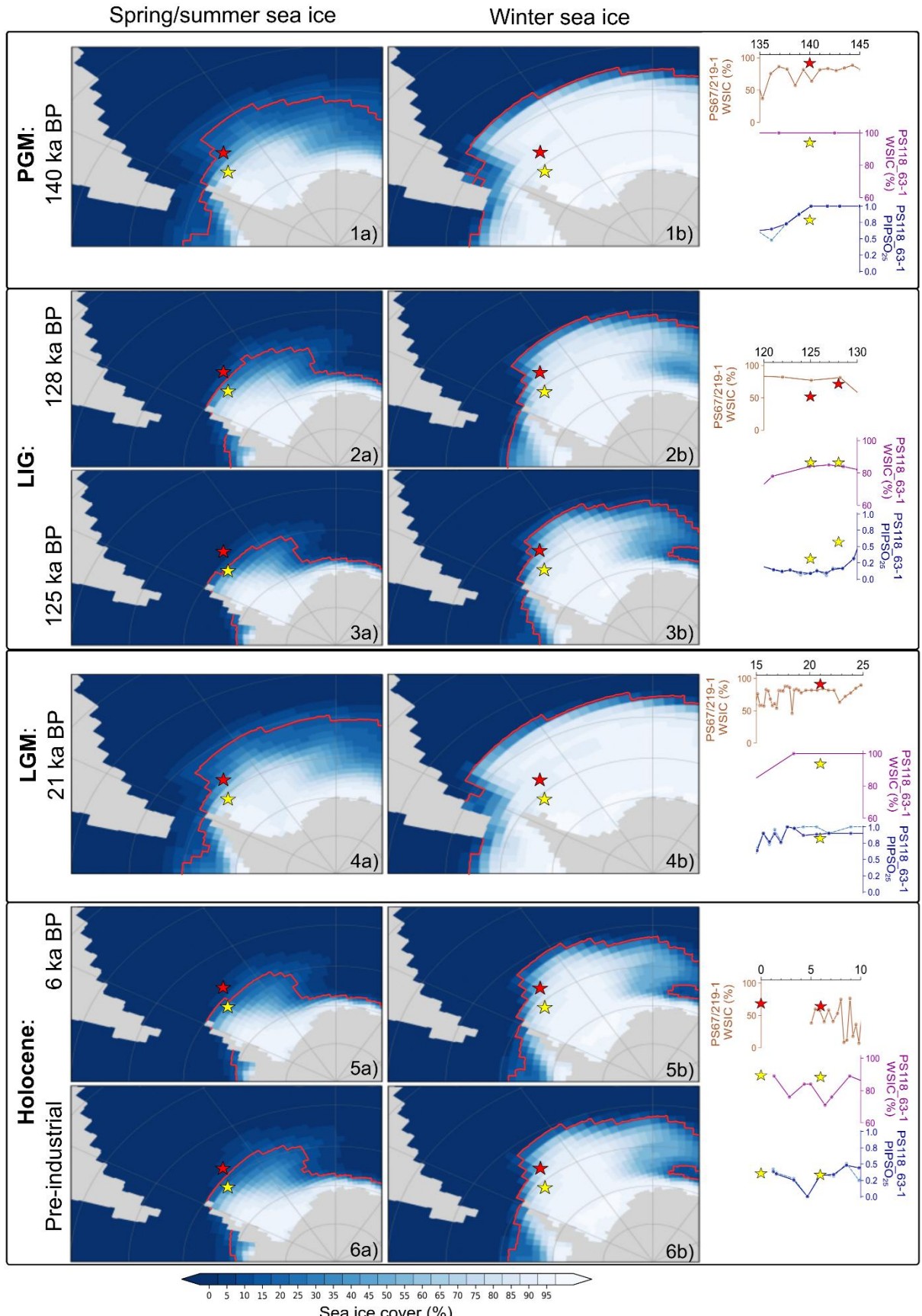

**Figure 5. Model-simulated mean a) spring/summer (NDJFMA) and b) winter (ASO) sea-ice cover for the various time slices: 1) PGM: 140 ka BP, 2) LIG: 128 ka BP, 3) LIG: 125 ka BP, 4) LGM: 21 ka BP, 5) mid-Holocene: 6 ka BP and 6) Pre-industrial. The red line depicts the sea-ice extent and is defined as the isoline**

 **of 15% sea ice coverage. Locations of marine sediment cores are indicated by stars: PS118_63-1 (yellow)**
**518** **and PS67/219-1 (red). Proxy-derived winter sea-ice concentration (WSIC) and spring/summer sea ice**
**519** **(PIPSO₂₅) at each core location are shown in the right-most panel. Additionally, model-simulated sea-ice**
**520** **values at each core site (yellow and red stars) for each time slice are plotted alongside the proxy data for**
**521** **comparison.**

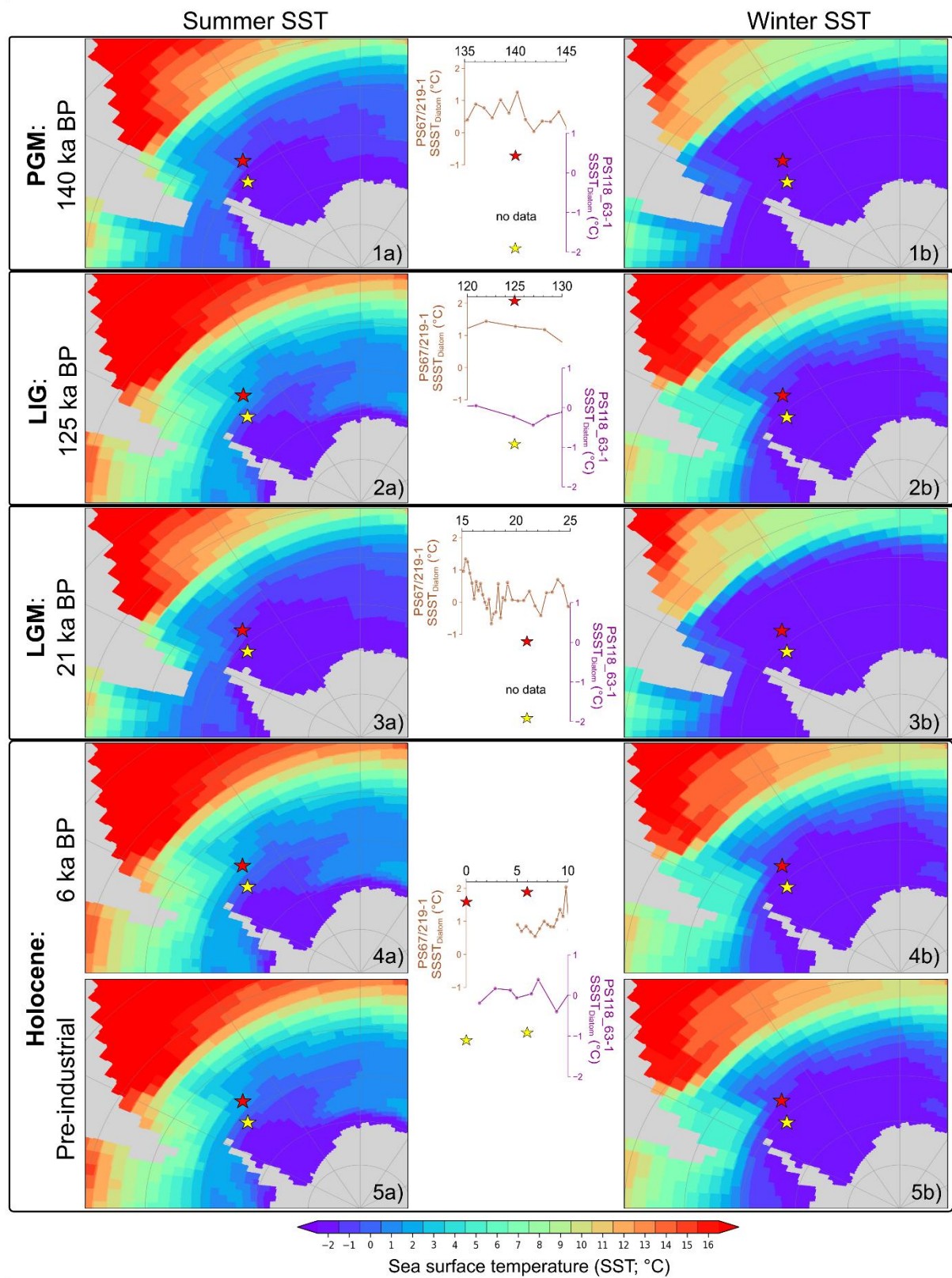

**523** **Figure 6. Model-simulated mean a) summer (DJF) and b) winter (JJA) sea surface temperature (SST) for**
**524** **the various time slices: 1) PGM: 140 ka BP, 2) LIG: 125 ka BP, 3) LGM: 21 ka BP, 4) mid-Holocene: 6 ka BP**

**and 5) Pre-industrial. Marine sediment cores, PS118_63-1 (yellow) and PS67/219-1(red), are indicated by the colored stars. Diatom-based summer sea surface temperature (SSST$_{Diatom}$) at both core locations is presented in the middle panel. The corresponding model-simulated SST at each core site (yellow and red stars) for each time slice is displayed alongside the proxy data for comparison.**

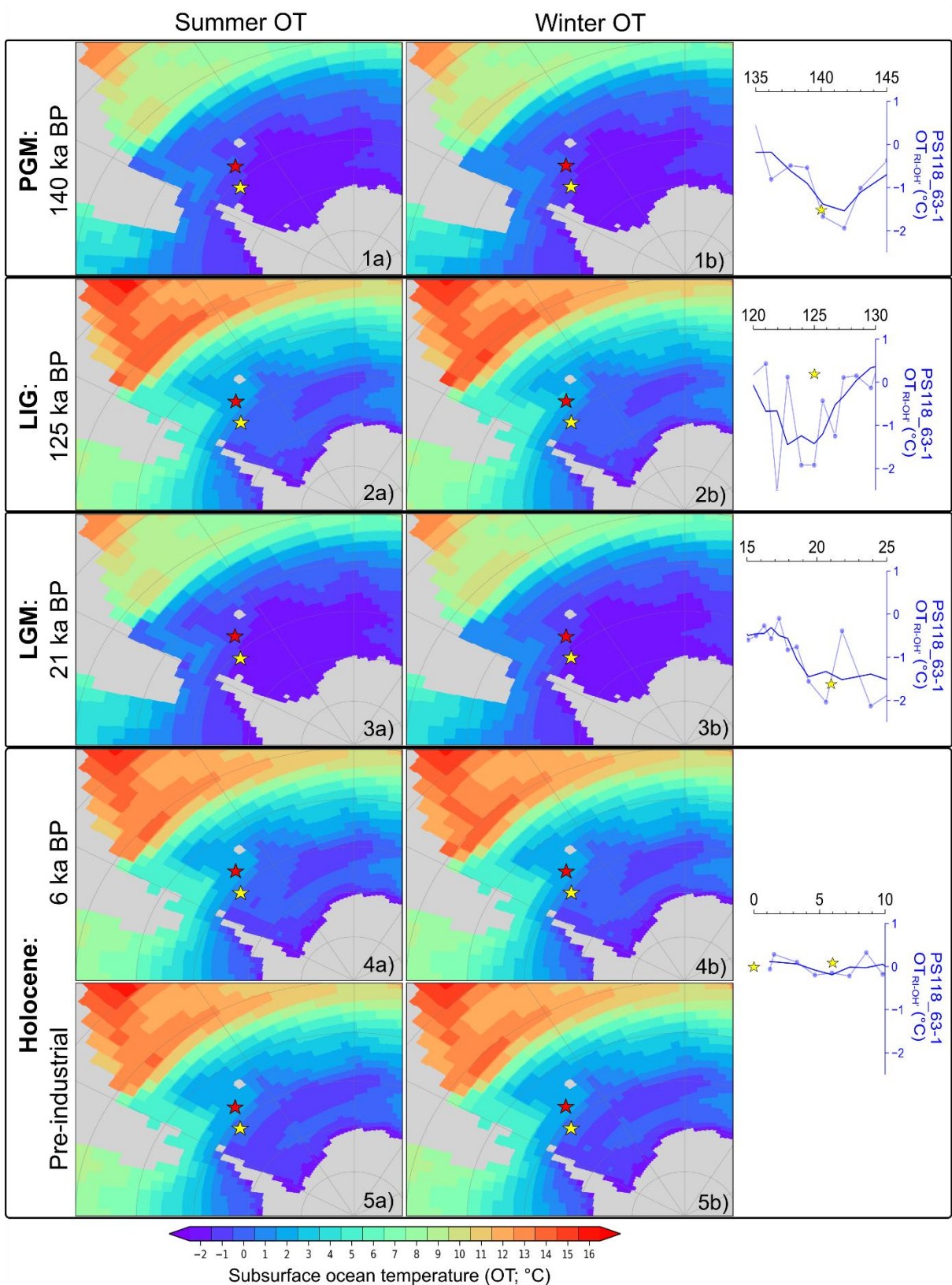

**Figure 7. Model-simulated mean a) summer (DJF) and b) winter (JJA) subsurface ocean temperature (OT; 220 m water depth) for the various time slices: 1) PGM: 140 ka BP, 2) LIG: 125 ka BP, 3) LGM: 21 ka BP, 4)**

For the SST and OT, the left columns (Fig. 6a and 7a) represent the summer (DJF) temperature, and the right columns (Fig. 6b and 7b) depict the winter (JJA) temperatures, respectively. The simulated-SST (Fig. 6) appears similar to that of the modeled sea-ice output. In general, widespread, low SST, close to the freezing point of seawater (that is approximately -1.9°C at salinity values modeled in the SO in our simulations), is exhibited across all time-slices during winter (Fig. 6b), while in summer (Fig. 6a), low SST mainly occurs in the Weddell Sea and along the coast of the Antarctic continent. For instance, at the core site PS118_63-1 in Powell Basin, Weddell Sea, there is no observed difference in SST between winter and summer during the glacial periods PGM (Fig. 6.1) and LGM (Fig. 6.3). Both sites were surrounded by sea ice during these periods (Fig. 5.1 and 5.4). However, in interglacials, a seasonal SST cycle of ~1°C is noted in the basin (Fig. 6.2, 6.4 and 6.5). In contrast, at the more northern core site PS67/219-1, the model estimates a seasonal SST cycle of ~1°C during the glacial periods (Fig. 6.1 and 6.3) and ~3.4°C during the interglacial (Fig. 6.2, 6.4 and 6.5). Moreover, the modeled climate states are characterized by spatial SST gradients between the two core locations of between 0°C (glacial) and ~0.4°C (interglacial) during winter. For summer SST, the gradient between the two core locations varies between ~1°C (glacial) and ~2.8°C (interglacial). As for the simulated OT, the model displays a ~1.6 and ~3°C glacial-interglacial variation at core sites PS 118_63-1 and PS67/219-1, respectively, but no appreciable OT change is observed between the winter and summer seasons of each time slices (Fig. 7). The model also reveals a spatial OT gradient between both marine core sites of ~0.7°C (glacial) and ~2.1°C (interglacial).

# 5 Discussion

## 5.1 Regional sea ice and oceanic conditions

### 5.1.1 *Penultimate Glacial Maximum – Termination II*

Our records show that during the PGM, the Powell Basin (core PS118_63-1) remained under a layer of persistent (sea) ice cover, as evidenced by a 100% WSIC and peak PIPSO$_{25}$ values inferred from the absence of diatoms, alongside notable reductions in IPSO$_{25}$ and HBI-triene concentrations (see also Sect 4.1 and 4.3). This coincided with the lowest levels of primary production reflected in the biogenic opal and TOC records (Fig 4b, c and e). This condition persisted until ca. 140 ka BP, when a decline in spring/summer sea ice (PIPSO$_{25}$) is observed, accompanied by a rise in TOC and subsurface ocean temperature (Fig. 4b, e and g). At a more northerly location in the South Scotia Sea, core PS67/219-1 records a less pronounced sea-ice cover during the PGM with WSIC fluctuating at around 65% and a 1-3% abundance of *F. obliquecostata* suggesting the proximity of a permanent sea-ice edge (Fig. 4d). These findings from the geological record are supported by our model simulation for the 140 ka BP time-slice, which shows an overall high simulated-WSI cover (94%; 92%), but slightly lower simulated-spring/summer sea-ice cover (79%; 27%) at core sites PS118_63-1 and PS67/219-1,

respectively (Fig. 5a). Likewise, higher $ssNa^+$ concentrations and $\delta^{18}O$ values from EDML ice core point
to cold conditions and an extensive sea-ice cover in the Atlantic region (Fig. 4a and j; EPICA Community
Members, 2006; Fischer et al., 2007).

Termination II (TII; 140-130 ka BP) marks the transition from a glacial into an interglacial
environment. The onset of this deglaciation was probably initiated by a warming event caused by a
maximum southern high latitude summer insolation at around 138 ka BP (Bianchi and Gersonde, 2002;
Broecker and Henderson, 1998) and further sustained by the Heinrich Stadial 11 (HS11) event
occurring in the Northern Hemisphere (NH) between 135 and 130 ka BP (Turney et al., 2020). The
HS11 is a prominent North Atlantic meltwater event that may have triggered the eventual shutdown of
the AMOC, thus reinforcing the warming in the SO via the bipolar seesaw effect (Marino et al., 2015).

In the Powell Basin, the WSIC remains high (100%) and only starts to decrease (80%) at ca. 134
ka BP, while gradually declining $PIPSO_{25}$ values since 140 ka BP accompany the onset of the
deglaciation and mark a shift from a perennial sea ice to a dynamic seasonal sea-ice cover (see Sect
4.5 for definition). A concurrent rise in subsurface ocean temperature is also observed during this
timeframe. In contrast, core PS67/219-1 in the South Scotia Sea recorded a different sea-ice regime
with generally lower and declining WSIC and <1% abundance of *F. obliquecostata,* suggesting a less
extended sea-ice cover. The different sea-ice conditions in both regions are supported by a higher
biogenic opal production recorded in the South Scotia Sea as compared to the minimum biogenic opal
content observed for the Powell Basin (Fig. 4e and f). The Powell Basin TOC profile is also different
from its opal counterpart, with the former peaking between 135-131 ka BP. We surmise that this peak
may relate to a preferential growth environment for non-siliceous marine organisms and/or increased
input of terrestrial organic matter during this interval.

The persistent warming was interrupted by a short period of spring/summer sea ice ($PIPSO_{25}$) re-
expansion and weakened decline in WSI towards the end of TII (ca. 132-130 ka BP; Fig 4b and c),
along with an increasing *Chaetoceros RS* abundance that peaks at ca. 131 ka BP (Fig. 3e). These
conditions coincide with the northward shift of the sea-ice edge at ODP Site 1094 around 129.5 ka BP
(Bianchi and Gersonde, 2002). A comparable reduction in SSST at around 131 ka BP is also observed
in the South Scotia Sea (core PS67/219-1, Fig. 4i) and apparent at ODP Site 1089 and core PS2821-1
(Cortese and Abelmann, 2002). In the Powell Basin, however, this cooling event is not reflected in
ocean temperature (Fig. 4g) and we propose that the lack of temperature change during this event may
be attributed to the discharge of meltwater from expanding sub-ice shelf cavities, which caused a
stronger stratification and an effective isolation of the warmer subsurface layer.
*5.1.2   Last Interglacial - MIS 5 stadials/interstadials*

Following the short-lived sea-ice expansion in Powell Basin at the end of TII, we observe a rapid
decline, and minimum spring/summer sea-ice cover is reached (see Sect 4.5) by ca. 129 ka BP (Fig.
4b). Lowest spring/summer sea ice ($PIPSO_{25}$) is observed between 126 and 124 ka BP, while minimum
WSIC is observed around 119 ka BP. These conditions promoted primary productivity, as reflected in
the maximum biogenic opal and TOC contents, at the respective timeframes (Fig. 4e). Likewise, sea
ice and temperature profiles from core PS67/219-1, the EDML ice core and model simulations also
favor a warm and predominantly open ocean condition for the South Atlantic sector throughout the LIG
(Fig. 4d, 4i, 5.3 and 6.3; EPICA Community Members, 2006; Fischer et al., 2007). Holloway et al.
(2017) investigated the simulated-spatial structure of the Antarctic WSI minimum at 128 ka BP with
respect to the $\delta^{18}O$-isotopic peak recorded in the East Antarctic ice cores. They tested numerous WSI
retreat scenarios and concluded that the $\delta^{18}O$ maximum could be explained by a significant decline in
Antarctic WSI, with the Atlantic sector experiencing the largest reduction of 67%. Contrastingly, while
our spring/summer sea ice (PIPSO$_{25}$) data aligns with their $\delta^{18}O$-accorded simulated-findings, our
diatom data - revealing a constant presence of WSI in the Powell Basin and South Scotia Sea with even
minor increases between 130 and 127 ka BP - disagrees. Furthermore, the WSI record from marine
core PS2305-6, located slightly north of our core site, also indicates the presence of WSI during MIS
5e (see also Supplementary Table S1 in Holloway et al., 2017; Bianchi and Gersonde, 2002; Gersonde
and Zielinski, 2000). We assume that the modeled winter sea-ice retreat seems to be valid for more
distal ocean areas, whereas at the core sites in Powell Basin and South Scotia Sea, ice-sheet-derived
meltwater may have acted as a driving mechanism fostering local sea-ice formation during winter, which
is not captured by the simulation in Holloway et al. (2017). Interestingly, the herein simulated sea ice at
the 128 ka BP time-slice corroborates our proxy-based data, indicating the presence of WSI in the
region amidst lower sea-ice concentration and continued retreat of sea ice over the spring/summer
seasons (Fig. 5.2). A similar sea-ice scenario is also established for the 125 ka BP time-slice,
considered to be the warmest period of the LIG (Fig. 5.3; Goelzer et al., 2016; Hoffman et al., 2017),
where Southern Hemisphere (SH) mid- to high-latitude spring insolation forcing reached a maximum
within the period from 130 ka BP to 125 ka BP (Lunt et al., 2013). The contrasting observation between
our marine sediment proxy and model data against that of the ice core $\delta^{18}O$-accorded simulated-finding
emphasizes the need for more robust marine-based reconstructions, especially south of the modern
sea-ice edge, to sufficiently substantiate model results for these regions, and to enable comprehensive
input knowledge for future model simulations and predictions (Holloway et al., 2017; Otto-Bliesner et
al., 2013).
The reconstructed SSST trends in the Powell Basin and the South Scotia Sea are largely
comparable with the atmospheric temperature profile from the EDML ice core (Fig. 4h-j), suggesting
atmosphere-ocean interactions in the study area. The lack of significant glacial-interglacial temperature
variability within the Powell Basin could potentially be linked to its locality and close proximity to the
continental margin, where constant mixing of cold ice-shelf water with the WDW persists. Within the
Powell Basin, both the SSST and subsurface ocean temperature started to decrease around 130 ka
BP. While the SSST appeared to have cooled from -0.2°C to -0.4°C (127 ka BP) and recovered
thereafter – similar to the dip observed in the EDML $\delta^{18}O$ profile – the subsurface ocean temperature
declined distinctly from 0 to ca. -1.9°C and remained cold until 124 ka BP (Fig. 4g and h). The variance
in the magnitude of decline observed between the two temperature records (SSST vs. OT) may be
attributed to the distinctly different seasonal signals depicted by the proxies (i.e., summer vs. annual
temperature) and water depths (0-10 m vs. 0-200 m; see also Sect 4.2 and 4.3). We speculate that the
decline in seawater temperature since 130 ka BP may be the result of intense melting of the Antarctic
ice sheet and sea ice, leading to a freshening of coastal waters. Similar to the modern-day Weddell
Gyre circulation (see Sect 2 for details), the increased discharge of cold (sea) ice-shelf meltwater into
the Powell Basin, via the Antarctic Coastal Current and Antarctic Slope Current, may have deepened
the cold-water stratification in the basin, thus causing the observed dip in ocean temperature between
130 and 124 ka BP. Turney et al. (2020) discovered that the WAIS had retreated from the Patriot Hills
blue ice area by the end of TII (130.1 ± 1.8 ka BP). This area is located 50 km inland from the present-
day grounding line of the Filchner-Ronne Ice Shelf. Their investigation revealed a 50 kyrs hiatus in the
blue ice record, indicative of a collapse of the ice shelf at the end of TII, followed by its subsequent
recovery during late MIS 5. Holloway et al. (2016) also propose a maximum ice-sheet retreat at around
126 ka BP based on distinct differences between the isotopic records observed for Mt Moulton and East
Antarctic ice cores. Assuming that the distinct reduction in spring/summer sea-ice recorded in core
PS118_63-1 was not confined to the Powell Basin but may reflect a more extensive sea ice decline in
the Weddell Sea embayment, we posit that this loss of sea ice (i.e., the loss of an effective buffer
protecting ice-shelf fronts) may have accelerated the disintegration of the Weddell Sea ice shelves and,
ultimately, the WAIS.
Following the peak of the LIG around 119 ka BP, the Powell Basin sea-ice records reflect a cycle
of sea ice advance and retreat throughout the remaining MIS 5 substages. WSIC strengthened and
remained at ca. 80%, while spring/summer sea ice (PIPSO$_{25}$) experienced a substantial increase
between MIS 5e and 5d (reaching PGM values at 5d), and remained elevated (> ca. 0.6) for the rest of
the MIS (Fig. 4b and c). This expansion of sea ice into MIS 5d, and its persisting presence throughout
the remaining MIS 5, is accompanied by a gradual decline in both sea surface and subsurface ocean
temperatures, along with reduced primary production. Likewise, an increasing WSIC, lowered SSST
and primary productivity are also noted in the South Scotia Sea (Fig. 4d-h). However, being more
northerly located, the South Scotia Sea experienced a lower and more varied WSIC (ca. 48 - 68%)
andminimum summer sea-ice cover evident by a lower abundance of *F. obliquecostata* (<1%) than in
the Powell Basin (Fig. 4d).
*5.1.3   Glacial period – Last Glacial Maximum – Termination I*
After MIS 5, Antarctica transited into the last glacial period (74-19 ka BP). In our Powell Basin
records, this is reflected in a northward expansion of the sea-ice extent (peak PIPSO$_{25}$ values and 100%
WSIC). Additionally, the lack of sea ice- and phytoplankton-related biomarkers and diatoms points
towards an extremely suppressed production in the basin (Fig. 3a and b, 4b and c). We postulate that
at that time the basin was likely covered by permanent sea-ice cover or a floating ice shelf, which
inhibited primary production in the underlying water column. The South Scotia Sea record (PS67/219-
1) further to the north also points to an overall higher winter and summer sea-ice cover, with elevated
abundance of *F. obliquecostata* (0 - 3%) during this period suggesting a permanent sea-ice edge close
to the core site (Xiao et al., 2016a). The oscillating patterns observed in both the sea-ice record and the
biogenic opal content further point to alternating advance and retreat phases of the sea-ice edge in the
South Scotia Sea (Fig. 4d and f; Allen et al., 2011).
In the Powell Basin, capped by an overlying (sea) ice cover throughout the glacial period,
subsurface ocean temperatures somewhat resemble the millennial-scale variability in the EDML
temperature profile (Fig. 4g). We presume that the subsurface temperature variations may possibly
reflect changes in the ocean circulation in the Atlantic sector of the SO (Böhm et al., 2015; Williams et
al., 2021). However, the age uncertainties and the low resolution of our subsurface ocean temperature
record hamper an affirmative conclusion, and more data points will be required to ascertain
corresponding oceanic variability.
The last glacial period culminated during the LGM between 26.5 and 19 ka BP with a most
northwardly extending sea-ice edge, as identified in several marine sediment cores (Fig. 4b and c;
Gersonde et al., 2005; Xiao et al., 2016a) and deduced from maximum ssNa$^+$ concentrations in the
EDML ice core (Fig. 4a; Fischer et al., 2007). Evidence from previous studies indicated the advance of
grounded ice sheet and island ice caps to the edge of the outer continental shelf (Davies et al., 2012;
Dickens et al., 2014). These grounded ice sheets were surrounded by floating ice shelves that extended
seaward to 58°S on the western side of Antarctica (Herron and Anderson, 1990; Johnson and Andrews,
1986). In the Atlantic sector, the 60 - 70% expansion of WSI towards the modern Polar Front (~50°S;
Gersonde et al., 2003) also promoted a northward shift of the summer sea-ice edge beyond core site
PS67/219-1 to around 55°S (Allen et al., 2011; Collins et al., 2012), which lead to restricted primary
productivity as reflected in the minimum biogenic opal content of core PS67/219-1 (Fig. 4f). The LGM
is also considered the coldest interval, with a northward expansion of the (sub)Antarctic cold waters by
4 - 5° in latitude towards the subtropical warm waters (Gersonde and Zielinski, 2000; Gersonde et al.,
2003). Sea-ice extent (Fig. 5.4) and SSST (Fig. 6.3) derived from our climate simulation during the peak
of LGM (21 ka BP) align with these findings. This distinct growth of the (sea) ice-field in the SO, coupled
with lower reconstructed and modeled LGM subsurface temperatures (Fig. 4g and 7.3), suggests an
intensified cold-water stratification at our core sites, and a possible northward displacement of the WDW
upwelling zone towards the edge of the summer sea-ice field (Ferrari et al., 2014).
TI began around 18 ka BP, when our records from Powell Basin indicate a transition from a
perennial-ice cover to a dynamic sea-ice scenario (see Sect 4.5), with several cycles of advance and
retreat. Similarly, the sea ice-related records from the South Scotia Sea (PS67/219-1) and the EDML
ssNa$^+$ record depict a decrease in sea-ice cover, along with rapid increases in primary productivity and
ocean temperature (Fig. 4). This deglaciation is attributed to a weakening AMOC circulation as a result
of reduced NADW formation caused by increasing NH summer insolation and significant ice sheet melt
at 18 ka BP, also known as Heinrich Stadial 1 (Clark et al., 2020; Denton et al., 2010; Waelbroeck et
al., 2011). The gradual warming of TI was interrupted by a brief cooling between 14 and 12 ka BP.
During this interval, our records reveal a short-term re-advancement in sea ice, coupled with a drop in
productivity and temperature (Fig. 4). This event seems to coincide with multiple South Atlantic records
(Xiao et al., 2016a) and higher ssNa$^+$ concentrations and a plateau in $\delta^{18}O$ values recorded in the EDML
ice core (Fischer et al., 2007). We hence propose this event to be the Antarctic Cold Reversal (ACR),
which is linked to the Bølling-Allerød warm interval in the NH via the bipolar seesaw mechanism (Pedro
et al., 2011; 2016).

   *5.1.4   Holocene*

Following the brief cooling of the ACR, the deglacial warming resumed its pace and Antarctica
transited into the present interglacial (Holocene: 12 ka BP-present), which is marked by intervals of
warming and cooling events (Bentley et al., 2009; Bianchi and Gersonde, 2004; Xiao et al., 2016a). Our
data support these findings and document periods characterized by seasonal/dynamic and minimum
sea-ice cover (see Sect 4.5) since 12 ka BP. We acknowledge that the age constraints and data
availability of core PS118_63-1 for the Holocene is limited and exercise caution on the interpretation of
the Holocene proxy records. Nevertheless, our data still permit the discrimination of Holocene warming
and cooling trends.
The Powell Basin experienced an overall rapid decline in the winter and spring/summer sea-ice
(Fig. 4b and c), concurrent with a rise in SSST (-0.5 to 0.5°C; Fig. 4h) and primary productivity between
12 and 5 ka PB (Fig. 4e), suggesting a seasonal sea-ice cover. The significant reduction in the
abundance of the *F. curta* gp (below 3%), WSIC and spring/summer sea ice (PIPSO$_{25}$; Fig. 4b and c)
culminates at ca. 5 ka BP and is accompanied by an elevated primary productivity reflected in rising
biogenic opal and TOC contents, which seems to indicate a brief open-ocean setting for the Powell
Basin during this warm interval. We further note fluctuating SSSTs, while the subsurface ocean
temperature remains relatively stable between 9 and 5 ka BP and the remainder of the Holocene (Fig.
4g and h). This somehow contrasts with a subtle decline in SSSTs recorded in core PS67/219-1 (Fig.
4i) in the South Scotia Sea, substantiated by the elevated presence of *Chaetoceros* rs recorded in core
PS118_63-1 (Fig. 3e). We may attribute this cooling to a northward export of increased glacial
meltwater. Our model simulation at 6 ka BP depicts a somewhat similar oceanic condition, with <40%
spring/summer sea ice at the studied sites (Fig. 5.5a). However, in comparison with our proxy records,
the model appears to have overestimated the WSI, SST and OT (Fig. 5.5b, 6.4 and 7.4). This
overestimation may be attributed to the complex ice-ocean interactions and feedbacks along the
Antarctic coastal region, which may not be fully represented in the model that has a spatial resolution
in the order of tens of kilometers and does not reflect any ice sheet dynamics.
While the limited age constraints for the Holocene in core PS118_63-1, preclude us from further
allocating short-term climate variations, we propose that the interval around 5 ka BP may reflect the
Holocene climate optimum, while the upper part of the core depicts the later Holocene conditions. Here,
increasing PIPSO$_{25}$ values and WSI reflect a re-expansion of seasonal sea ice still permitting primary
productivity as derived from elevated biogenic opal and TOC contents (Fig. 4b, c and e). The climate
optimum experienced in the Powell Basin seems to correspond to the mid-Holocene climate optimum
identified in sediment cores from the South Orkney Plateau between 8.2 and 4.8 ka BP and around
Antarctica (Crosta et al., 2008; Denis et al., 2010; Kim et al., 2012; Lee et al., 2010; Taylor et al., 2001).
However, reports of differing timings and mode for the mid-Holocene climate optimum around the
Antarctic Peninsula have been noted in previous studies (Bentley et al., 2009; Davies et al., 2012;
Shevenell et al., 1996; Taylor and Sjunneskog, 2002). Vorrath et al. (2023) determined the mid-
Holocene climate optimum to have occurred between 8.2 and 4.2 ka BP, based on biomarker analyses
of a sediment core from the eastern Bransfield Strait. They suggest that the climatic changes at their
core site were influenced predominantly by the warm Antarctic Circumpolar Current rather than the
cold-water Weddell Sea. This is contrary to a shorter climate optimum (6.8-5.9 ka BP) proposed by
Heroy et al. (2008), where they examined the climate history of western Bransfield Strait using sediment
and diatom analyses. Such diverse research outcomes highlight the complexity of responses to micro-
region variations in glacial, atmospheric and oceanic changes in the Antarctic Peninsula throughout the
Holocene (Bentley et al., 2009; Davies et al., 2012; Heroy et al., 2008; Vorrath et al., 2023).
**5.2    Comparison between interglacials / transition periods**

A comparison of the environmental changes caused by climate warming during TII and TI as well
as the peak LIG and the Holocene, may yield valuable information on common or different driving and
feedback mechanisms. As marine cores PS118_63-1 and PS67/219-1 provide continuous records of
the environmental evolution in the northwestern Weddell Sea and South Scotia Sea, respectively, dating
back to at least 145 ka BP, they offer a distinct opportunity to evaluate (sea-ice) conditions between the
two terminations (TII and TI) and both warm periods (LIG and Holocene), particularly in proximity to the
continental margin. Denton et al. (2010) studied the last four terminations and concluded that the
terminations were triggered by a sequence of comparable events: maximum NH summer insolation that
caused substantial NH ice sheet melting (due to marine ice sheet instability) over an extended (>5 kyrs)
NH stadial interval. The huge release of meltwater slowed the AMOC, thus triggering an intense
warming in the southern high-latitudes through the bipolar seesaw teleconnection, accompanied by a
poleward shift in the southern westerlies. In line with this hypothesis, our records from cores PS118_63-
1 and PS67/219-1 portray a consistent and rapid decline in sea ice throughout both terminations (TII
and TI). Interestingly, both deglaciations feature a short-term readvance of sea ice during their latest
stage, at ca. 130 ka BP and during the ACR, respectively, likely due to meltwater-discharge from
retreating ice shelves/ice sheets in the SO. This suggests that short-term sea ice growth stimulated by
deglacial meltwater may be a common feature during glacial terminations. Despite commonalities in the
sea-ice records, some differences are discernible. For instance, during TII, there is an abrupt surge in
biogenic opal in the South Scotia Sea, along a consistent rise in TOC content within the Powell Basin.
In contrast, TI exhibits a pattern characterized by a gradual increase with periodic fluctuations
throughout the termination for both TOC and biogenic opal content. Additionally, the South Scotia Sea
(PS67/219-1) recorded a higher mean biogenic opal content and SSST across TII (35%; 0.7°C) than TI
(26%; 0.5°C). Likewise, in the Powell Basin (PS118_63-1), higher mean TOC and subsurface ocean
temperature are perceived during TII (0.5%; 0°C) than during TI (0.4%; -0.3°C). These data are in
agreement with the EDML $\delta^{18}O$ record, which registered a stronger deglacial amplitude (32%) in TII
than TI (Masson-Delmotte et al., 2011). Broecker and Henderson (1998) also speculated that the
amplitude of the SH summer insolation during TII was higher than during TI. Additionally, a delay of
approximately 10 kyrs between the SH and NH summer insolation (and subsequent NH ice sheet
melting) during TII – as compared to TI's SH summer insolation peak just before the melting of the NH
ice sheet – probably contributed to a more pronounced TII warming in the SO. The differing magnitude
of warming observed between both core sites in the South Atlantic, however, is likely attributed to their
latitudinal differences.
The climate during the LIG appeared to be warmer than during the Holocene. In the Powell Basin,
the LIG peak interval (i.e., MIS 5e) was characterized by a significantly reduced spring/summer sea-ice
cover and peak productivity, while a higher spring/summer sea-ice cover, along with an only gradually
increasing productivity are observed for the Holocene warm period (Fig. 4b and e). However, no
significant difference in the WSIC between both interglacial was noted. The discrepancy in warming
intensity likely occurred seasonally and coincided with maximum summer insolation (see also Fig. 4 in
Bova et al., 2021). Nonetheless, a lower mean annual regional insolation (-1.1 W/m$^2$ difference; Laskar
et al., 2004) during the LIG does not explain the warmer conditions observed in the region. Bova et al.
(2021) hypothesized that the LIG was relatively warmer than the Holocene as a result of its preceding
deglacial dynamics: specifically, the magnitude of the last deglaciation was half that of the penultimate
deglaciation – where a rapid and intense warming destabilized and significantly reduced the (sea) ice
cover to near modern-day level by the onset of the LIG (Bova et al., 2021), and possibly a collapse of
the WAIS in the first half of the LIG (Pollard and Deconto, 2009; Sutter et al., 2016). As such, we opine
that the lower magnitude of warming during TI was a consequence of spatially and temporally varying
retreats and advances in ice cover (including sea ice, ice shelves and glaciers) in the SO. The higher
ice coverage throughout the Holocene resulted in a higher surface albedo and a cooler Holocene, as
compared to the LIG. This is witnessed in our rather variable Holocene sea-ice proxy records (Fig. 4b
and c) and differing reports of mid-Holocene warming and repeated fluctuations in environmental
conditions around Antarctica (see sect 5.1.4; Bentley et al., 2014; Davies et al., 2012; Ó Cofaigh et al.,
2014).

**5.3  Evaluating COSMOS performance: Addressing boundary conditions and model selection**
With regard to COSMOS simulations, we note very similar sea-ice conditions being depicted for
the peak interglacial 125 ka BP and 6 ka BP time slices (Fig 5.3 and 5.5), while subtle differences are
resolved for SSTs and OTs (Fig. 6.2 and 6.4, 7.2 and 7.4, respectively). When considering the disparity
observed in our proxy data between these two interglacial intervals, we infer that these similarities in
the simulations likely result from using the same geographic boundary conditions for both time slices,
while climate forcing data (e.g., greenhouse gases, orbital parameters) differ, of course. Our study
aligns with the PMIP framework in maintaining a constant modern-day geography across each
interglacial time slice, specifically the mid-Holocene (e.g., 6 ka BP) and the LIG (e.g., 128 and 125 ka
BP). For the 6 ka BP time slice, this decision is supported by evidence indicating that ice sheets had
reached their modern configuration (Otto-Bliesner et al., 2017). In the case of the LIG, the use of the
modern ice-sheet configuration is primarily due to uncertainties in the LIG reconstructions (Otto-Bliesner
et al., 2017). We acknowledge that the consideration of a single geographic configuration throughout
the LIG certainly is a simplification. However, it is also important to note that the changes in the Antarctic
ice sheets' contribution to global mean sea level were small between 128 and 125 ka BP, compared to
the remainder of the LIG (Barnett et al., 2023). Therefore, we propose that using a constant ice-sheet
configuration for our LIG time slices is a reasonable approximation – in particular when we consider the
lack of robust alternative ice sheet configurations that could have been used as a boundary condition
for the climate model. Similarly, we estimated a constant ice-sheet setting for both the PGM and LGM
time slices. While there are indications of different NH ice- sheet extents between the two glacial periods
(Rohling et al., 2017), uncertainty remains regarding the exact distribution of ice on Antarctica.
Understanding this distribution is crucial to determine whether different ice-sheet configurations should
be considered for the boundary conditions of the respective glacial climate simulations. Given the varied
trends observed in our proxy data for each glacial and interglacial periods, we propose that future
studies should explore different plausible Antarctic ice-sheet configurations and their effects on glacial-
interglacial sea ice and oceanic conditions in the SO, particularly in the coastal regions.
In our modeling approach, we have relied exclusively on simulations from COSMOS rather than
adopting a multi-model approach based on available PMIP simulations. This decision was motivated by
the need to cover specific time slices pertinent to our study (see also Sect 3.5). To validate the reliability
of our results, we conducted a comparison of COSMOS-simulated sea-ice cover and SST results
against those from the PMIP3 and PMIP4 ensemble models. We refer to Supplement S3.4 for full detail.
In general, the model-to-model comparison shows good agreement (<2σ threshold) between our
COSMOS results and those from the PMIP3 ensemble – especially at our study locations, with some
disagreement noted for the 21 ka BP time slice (Supplementary Fig. S4 and S5, S8 and S9). These
deviations largely occur around the sea-ice edge and are primarily due to uncertainties generated within
the PMIP3 ensemble itself. In contrast, our COSMOS-to-PMIP4 ensemble comparison shows greater
disagreement. The COSMOS simulation shows a milder warm bias in the SO compared to various other
PMIP3 models (Lunt et al., 2013), whereas CMIP6 models, which provide the foundation for PMIP4,
are documented to have a warm bias in the SO (Luo et al., 2023). Beyond the difference in warm bias,
the disagreements between COSMOS and PMIP4 may arise from several factors, including evolution
of modeling protocols, boundary conditions, and model development from PMIP3 to PMIP4, with
COSMOS remaining a PMIP3-class model. Based on the comparative outcomes, we demonstrate that
our results align with PMIP in many relevant aspects, though this comparison is limited by the
incomplete coverage of time slices within PMIP. Where our model shows disagreement with the PMIP3
ensemble, the uncertainty within the ensemble itself is quite large. This highlights that the uncertainty
in simulated sea-ice conditions at our core locations, which we acknowledge as a limitation of using
only one model in our study, is not necessarily mitigated by using an ensemble of models instead. Given
that COSMOS is mostly within the 2σ threshold – defined as a measure for agreement with the PMIP3
ensemble – at the study sites, we would not expect to derive substantially different inferences if we
relied on the PMIP3 ensemble instead. Although COSMOS has not undergone the updates that PMIP4
models received and has been exposed to boundary conditions only partly comparable to PMIP4
simulations, to date it remains one of the most extensively utilized models for reconstructing Quaternary
climates and beyond. This enables our study's results to be considered within the much larger context
of the Cenozoic climate. Despite the aforementioned limitations, it is worth noting that COSMOS has
been successfully employed alongside other PMIP4 models (Stepanek et al., 2020).

## 6   Summary and conclusions

Multiproxy analyses on marine sediment core PS118_63-1 from the Powell Basin provide new insights into the glacial-interglacial environmental variability in proximity to the Antarctic continental margin. With the use of the novel sea ice and open-water biomarkers and diatom assemblage data, alongside primary productivity proxies, we are able to reconstruct sea-ice conditions in the Powell Basin for the past ca. 145 kyrs. Our findings reveal year-round ice-cover with minimal productivity during glacial periods, while dynamic sea-ice conditions with varied productivity are recorded in the Powell Basin during climate transitions and interglacial periods, such as the Holocene and MIS 5. Peak reduction in sea ice and near open ocean conditions are noted for MIS 5e. In contrast, no significant glacial-interglacial temperature variation was registered in the basin, which is attributed to the cold-water regime of the Weddell Sea. Comparison between the current and last interglacial, and their respective climate transitions (TI and TII), suggests a relationship between deglacial amplitude and warming intensity during the corresponding interglacial: in general, an abrupt and intense (gradual and slow) deglaciation leads to a warmer (cooler) interglacial, with higher (lesser) ice-sheet retreat (Bova et al., 2021). Our data presented in this study reinforce earlier paleo sea-ice reconstructions in the South Atlantic sector of the SO and provide new insights into the ice-proximal sea-ice response during varying climate conditions. Evaluation of both proxy and model data highlights similarities between sea-ice reconstruction and simulation. However, notable discrepancies remain, such as the differing proxy-model data observed for the Holocene compared to the LIG, and subsurface temperature profile for the LIG. It is therefore pivotal to explore different Antarctic ice-sheet configurations in future studies, as well to expand on the paleoclimate data for the region. These will help to close the gap in our understanding of ocean-ice-atmosphere interactions and dynamics and ultimately enhance climate model predictions closer to the Antarctic continental margins.

**Data availability.** Proxy data mentioned in this article will be available at https://doi.org/10.1594/PANGAEA.965042 (Khoo et al., 2024), and COSMOS model output will be accessible at https://doi.org/10.1594/PANGAEA.972654 (Stepanek et al., 2024). For specific model output requests beyond the climate variables included in the PANGAEA data publication, please contact Christian Stepanek at christian.stepanek@awi.de. CMIP/PMIP data is available via the Earth System Grid Federation using one of their publicly available data portals (e.g., https://esgf-data.dkrz.de/search/cmip5-dkrz/ and https://esgf-data.dkrz.de/search/cmip6-dkrz/).

**Code availability**. Requests for the source code of the COSMOS climate model should be directed to the Max Planck Institute for Meteorology, Bundesstrasse 53, 20146 Hamburg, Germany.

**Supplement.** The supplement related to this article is available online at:

**Author contributions.** This study was conceived by WWK and JM. Data collection and interpretation was conducted by WWK, together with OE (diatom), JM (HBI), JH and GM (GDGT). WG produced the U/Th-dating data. CS and GL selected, documented, and post-processed the data from an ensemble

of simulations that provided the climate model data for this study. Three of the six simulations presented here, namely *lig125k*, *lig128k*, and *pgm140k*, represent previously unpublished climate model output created by PG. WX supplied unpublished data for PS67/219-1. WWK wrote the paper and created the visualizations, supported by CS who visualized model output and interpolated climate model output to core locations. JM supervised the study. All authors contributed to the analyses, discussion of the results, and the conclusion of this study.

**Competing interests.** The authors declare that they have no conflict of interest.

**Acknowledgements.** We thank the captain, crew and science team of the RV Polarstern cruise PS118 (Grant No. AWI_PS118_04). Special thanks go to Michael Schreck, Nele Steinberg, Sabine Hanisch and Frank Niessen for PS118 marine geology operations. Appreciation is also extended to Denise Diekstall (HBI), Mandy Kuck (HBI), Ulrike Böttjer (Biogenic Opal) for their support. Simon Belt is acknowledged for providing the 7-HND internal standard for HBI quantification. This research is funded through the Alfred Wegener Institute Helmholtz Centre for Polar and Marine Research (International Science Program for Integrative Research in Earth Systems, INSPIRES II). Gerrit Lohmann, Paul Gierz, and Christian Stepanek are funded through the Alfred Wegener Institute's research program: Changing Earth - Sustaining our Future of the Helmholtz Association. Christian Stepanek also acknowledges funding from the Helmholtz Climate Initiative REKLIM. We acknowledge the World Climate Research Programme's Working Group on Coupled Modeling for CMIP, and the Paleoclimate Model Intercomparison Project and its working groups for coordinating the model intercomparison in PMIP3 and PMIP4. Appreciation is extended to the climate modeling groups (listed in Table S4) for their contribution and availability of model output to CMIP5/6 and PMIP3/4. The U.S. Department of Energy's Program for Climate Model Diagnosis and Intercomparison is recognized for providing coordinating support and leading software infrastructure development with the Global Organization for Earth System Science Portals. The Earth System Grid Federation is also acknowledged for preserving and providing CMIP and PMIP model output. We are also appreciative of the support from the Alfred Wegener Institute's Open Access Publication Funds. Lastly, we thank the editor, Dr. Alberto Reyes, Dr. Xavier Crosta and an anonymous reviewer for their constructive comments that helped to improve the paper.

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
