# Peer review of "Ice-proximal sea-ice reconstruction in Powell Basin,"

_EGUsphere, 2024_

## Author Comment (AC1)

**Response to Referee #2**

We sincerely thank referee #2 for your insight and constructive comments and recommendations on our original submission. All comments have been carefully reviewed, and integrated into the revision to enhance clarity and refinement of the manuscript. Below, please find our responses (in red; *revisions already made in the manuscript are shown in italics*), to your comments.

**RC2.1.1 General comments:** One of my biggest concerns is, that Khoo et al. focus in their Introduction on the active role of sea ice in the climate system. Further, they mention to identify potential tipping points in the ice-ocean-atmosphere system by reconstructing past sea-ice changes. In the Discussion, the authors exclusively discuss the reaction of sea ice to meltwater or solar forcing. Instead of discussing the active role of sea ice in the climate system and potential tipping points, they discuss the forcing mechanisms on sea ice. Which is, nevertheless, extremely important to understand.
In the Abstract, sea ice-glacier interactions are put into focus, which is only shortly mentioned in the Introduction. In the Discussion this process is only mentioned in Chapter 5.2. I have the feeling the Authors could be more precise here and try to set the focus of this study more clearly.

Author's response: Thank you for your comment. Our intent of the first paragraph in the Introduction is to emphasize the importance of studying (past) sea ice variability, hence we will retain the impact of sea ice on the climate system and glacier-sea ice interactions. However, we will remove "identifying potential tipping points in the ice-ocean-atmosphere system" to avoid confusion. Regarding the comment on sea ice-glacier interactions, we refer to the response provided for RC2.3.1 in the Specific comments below.

**RC2.1.2 General comments:** I noticed, that a lot of abbreviations are used, which often are not necessary because terms are not used regularly throughout the manuscript. This makes the manuscript hard to read: e.g. HSSW only used 3 times, SOM only used once, HASO also only used once. I would recommend to only use abbreviation if a term is used more than 3 times.

Author's response: Thank you for your suggestions. We will update the revised manuscript accordingly.

**Specific comments**

**RC2.2 Abstract:**
- A lot of detail is given on the used proxies, the model study, however, is not mentioned.
- The general outcome of the study is very short. I would appreciate a bit more detail.

Author's response: We agree with the reviewer, and will add the use of the numerical model as well as expand on the outcome of this study to the revised abstract.

**RC2.3.1 Introduction:** In the Abstract you put the focus on sea ice-glacier interactions, which is also discussed later in the Discussion. However, in the Introduction, this is only mentioned in one sentence, and the focus is laid on the feedback mechanisms of sea ice on solar radiation and ocean circulation. More information on the glacier-sea ice interaction specifically for Antarctica would be nice.

Author's response: We agree with the reviewer and provided examples of ice shelf-sea ice studies in Antarctica to the revised manuscript. The following has been added to the revised manuscript:

*"Sea ice also serves as a crucial buttressing force at the ice front, effectively preventing or delaying the occurrence of potential calving events (Robel, 2017). This phenomenon was evident at locations such as the Mertz Glacier Tongue (Massom et al., 2015) and the Totten Ice Shelf (Greene et al., 2018) in East Antarctica. Furthermore, the presence of a sea-ice buffer in front of the ice terminus acts to diminish ocean swells as they propagate towards land. For instance, Massom et al. (2018) observed a substantial increase (orders of magnitude) in wave energy experienced at the fronts of the Larsen ice shelves and the Wilkins Ice Shelf when the sea-ice buffer was removed."*

**RC2.3.2 Introduction (L48-54):** Jumping between proxy archives here, which is very confusing. Please separate sedimentary and glacial proxies.

Author's response: We acknowledge the recommendation and incorporated the changes into the revised manuscript:

*"Presently, numerous methods are used to reconstruct past sea-ice conditions, including biogenic proxies (e.g., biomarkers, diatoms, dinoflagellate cysts, foraminifera and ostracods) and sedimentological proxies (e.g. ice-rafted debris) in marine sediments, as well as chemical compounds archived in ice cores (e.g., methanesulfonic acid and sea-salt (ssNa+); de Vernal et al., 2013 and references therein)."*

**RC2.3.3 Introduction (L56-64):** I agree with your statement, that the number of LIG sea ice reconstructions are limited, However, in your text you mention 184 studies in sea ice in Antarctica summarized in Crosta et al., (2022). This is in strong contrast to the general phasing you use, as reconstructions being "limited" and scares". Hence, I would recommend to change the wording in L56 and L64, to point out that biomarker studies (with their advantages over other proxies) are few in Antarctica.

Author's response: We agree with the reviewer that the number '184' is considered strong. However, the intent of this statement(s) is to highlight the disparity in number of (paleo)records between (a) different regions in the SO (i.e. opal belt (40 – 60ºS) vs. south of 60 ºS), (b) different timescales (Holocene vs. LGM vs. LIG and older), and (c) lack of records from past warmer periods (i.e. interglacials). Rather than the number of diatoms vs. biomarker studies in the SO. Hence, we will maintain the original statement, with some revision for clarity:

*"The compilation documents 20 studies on sea-ice variability during the Holocene (0-12 ka before present (BP)), 150 records detailing changes at the Last Glacial Maximum (LGM; ca. 21 ka BP or Marine Isotope Stage (MIS) 2), and a mere 14 sea-ice records dating back to around 130 ka BP. Notably, just two records extend beyond MIS 6 (ca. 191 ka BP; see also Fig. 3 in Crosta et al., 2022). Their work underscores the pronounced dearth of (paleo) sea-ice reconstructions, particularly in regions south of 60°S, notably in the Atlantic sector, and during the Last Interglacial (LIG) and beyond. This scarcity of records, in particular proximal to the continental margin, is attributable to difficulties in recovering marine sediment cores in the polar regions that at present are still subject to heavy year-round ice cover, and a lack of continuous sedimentary records due to erosion and disturbance at the sea floor during past glaciations."*

**RC2.3.4 Introduction (L117-125):** I would recommend to formulate the research question you aim to answer more clearly. Here you mention to close a knowledge gap, which I feel is not sufficient enough and not doing right by the relevance of your study.

Author's response: Thank you for the comment. We agree that the term "fill this gap" may not be the right term to use in this case. We will therefore replace 'fill this gap' with 'aim' to better express our intent of this study.

**RC2.4 Results (L320):** input instead of inputs

Author's response: Thank you for your comment. We corrected this.

**RC2.5.1 Discussion (L525-528):** Could you elaborate more on the lack of SSST and OT reduction at your core site in Powell basin. How do you explain this while associating it with increased meltwater inflow from the Antarctic Ice Sheet?

Author's response: Thank you for the comment. We will add the following statement to address the concern:

*"In the Powell Basin, however, this cooling event is not reflected in ocean temperature (Fig. 4g) and we propose that the lack of temperature change during this event may be attributed to the discharge of meltwater from expanding sub-ice shelf cavities, which caused a stronger stratification and an effective isolation of the warmer subsurface layer."*

**RC2.5.2 Discussion (L539-545):** How do you explain the strong seasonality in sea ice concentrations in Powell Basin?

Author's response: Thank you for the comment. As presence of WSI is also indicated in marine core PS2305-6 (Bianchi and Gersonde, 2002; Gersonde and Zielinski, 2000), located slightly north of our

core site, we propose that ice-sheet derived meltwater may have acted as a driving mechanism in promoting sea ice formation in Powell Basin during winter. We will incorporate this discussion in the revised manuscript.

**RC2.5.3 Discussion (L657-702):** I appreciate the acknowledgement of the large age uncertainties for the Holocene, however, the low data availability in your record should also be acknowledged. The interpretation of warm/cold or more/less sea ice phases of the Holocene is based on one data point only. I am not sure if it is wise to interpret these small-scale Holocene changes in your record. I would rather focus on the general glacial-interglacial trends, which is the focus of your study and the strength of your records. At least be more careful in the Holocene section of your Discussion.

Author's response: We agree with the reviewer that the data used in the interpretation of the climate variability of the Holocene is limited. However, we believe that, with the analysis of multiple proxy data, the interpreted climate variability (i.e. warm/cold intervals) remains valid. Nevertheless, we have revised the text to point out the low data availability for the interpretation of the Holocene climate changes.

*"We acknowledge that the age constraints and data availability of core PS118_63-1 for the Holocene is limited and exercise caution on the interpretation of the Holocene proxy records. Nevertheless, our data still permit the discrimination of Holocene warming and cooling trends."*

**RC2.6 Figures:**
Fig 1
● The insert map should at least include an overview circulation and regional names, e.g. Scotia Sea, otherwise it is hard to follow Chapter 2. Maybe a map showing the Atlantic Sector with regional names, currents, etc. would be more sufficient.
● The dashed light blue line (summer sea ice extent) is barely visible

Fig 4
● This figure is the key figure of the manuscript, but hard to decipher as it holds a lot of records and data. Please add numbers, letters, etc. to refer to the single plots in the figure captions. I see that the authors try to establish a color-coding distinguishing between different core locations (PS62/219-1 always in orange-brown colors). Maybe this could be done better. Further I am not sure if plotting the diatom species cugr. And F. obli on the same axis. Variations of F. obli are hardly visible.
● OTRI-OH' plot: Please indicate in captions what the light blue and dark blue (running average?) lines represent.

Fig 5
● The brown star (PS62/219-1) is hard to see with dark blue background. The red line (15% sea ice coverage) is hardly visible at all.

Fig 6
● I understand SSSTdiatom for PS118_63-1 is not available for the PGM and LGM, it is irritating to have an 'empty' graph. I would suggest to add an 'n.a.' onto the graph where data is not available
● The SST scale could be adjusted, as the largest change occurs within the SST range of -2 – 14 °C, if you adjust the SST scale the critical changes would stick out more.

Fig 7
● Here the OT scale should also be adjusted, shown OT stop around 15-16°C but the scale goes up to 21°C
● What does the dark blue line indicate? Running average of OT?

Author's response: We thank the reviewer for the detailed review to improve the figures. We will incorporate all suggested changes in the revised manuscript.

**RC2.7.1 Supplementary Material (Fig S1):**
● Could you please give more detail on the comparability of XRF Ti counts and EDML - d18
● I do not understand how you choose peaks for calibration in both records, and why you excluded two of them.

- Why do you use Ti counts alone? A more sophisticated approach would be to use element ratios? (Hennekam & deLange, 2012)

Hennekam, R., deLange, G. (2012). X-ray fluorescence core scanning of wet marine sediments: methods to improve quality and reproducibility of high-resolution paleoenvironmental records. Limnology and Oceanography: Methods 10

Author's response: Thank you for your comment. We will make adjustments to Supplement S1 to provide additional details on the selection of tie points used in the final age model. Pertaining to the use of Ti counts alone, we recognize that utilizing XRF-element ratios could offer a more nuanced analysis. However, in our case, we believe that the use of Ti counts alone is sufficient to capture the terrigenous signal in our study. Moreover, we integrated other proxy records such as TOC, MS and wet bulk density for a comprehensive age-depth comparison with the EDML δ18O record. This information will be included in the revised manuscript. See also similar comment from reviewer #1 RC1.1.

**RC2.7.2 Supplementary Material (Fig S2b):**
- I find it hard to see a correlation pattern here. There should be more information on how the peaks where chosen (or not) for calibration).

Author's response: Thank you for your comment. We acknowledge that there appears to be minimal correlation evident in this XRF-Fe pairing. This is why a significant portion of the tie points associated with this pairing were eventually rejected. Nonetheless, we did not completely dismiss this pairing as we aim to establish a robust selection of tie points. During the analysis, we compared multiple records (from the same core) simultaneously to ensure that the age-depth ranges for the tie points do not differ too much between each record. More details on the selection/rejection of the tiepoints will be added to the revised manuscript to provide clarity (refer also to above response for RC2.7.1).

**RC2.7.3 Supplementary Material (Table S1):**
- What reservoir correction did you use?

Author's response: Thank you for your comment. The reservoir ages for the two radiocarbon dates are approximately 2.2 kyrs. They were derived using the PaleoDataView software and will be included in Supplementary Table S1.

---

## Author Comment (AC2)

**Response to Referee #1**
We express our gratitude to Dr Xavier Crosta (referee #1) for your insightful and constructive comments and suggestions on our original submission. All remarks have been thoroughly evaluated, and revisions will be integrated to enhance clarity and further refine the manuscript. Please find our responses to your comments, in red *(revisions already made in the manuscript are shown in italics)*, below.

**RC1.1 – Age model (section 3.1; Supp S1):** I appreciated the effort made by the author to develop a robust age model through radiocarbon dating and comparison of several proxies to the U1357 nearby site and EDML. I believe that the age model is as good as it can be for a core from this complicated region. However, I would have appreciated more information about (1) why the negative relationship between Ti and EDML d18O during the 150-27 kyrs BP shifts to a positive relationship at younger times (Fig S1); (2) why some of the most obvious tie-points were disregarded (Fig S2d for example); and (3) are the radiocarbon dates reliable? I also noted that the age model is developed by comparison to U1357 records. U1357 age model until MIS6 is based on the comparison of its MS record to EDC dust record (a different ice core than the one on which PS118 Ti record is here tuned) assuming that the MS signal is a direct proxy for dust deposition in the western South Atlantic, which is not completely true as magnetic particles and iron are transported from the Antarctic Peninsula. It is also said in the present manuscript that there are environmental and oceanographic differences between PS118 (Powell Basin) and PS67 (twin core of U1357; Scotia Sea), so I wonder if it is sensible to tune PS118 to U1357. I guess this is the best one can do, but an honest evaluation of the incertitude (x kyrs) must be provided. The corollary is that it might prove difficult to interpret millennial changes and accurate timing of rapid changes, especially between different cores (PS118 vs PS67).

Author's response: We agree with the reviewer that the development of the age model for core PS118_63-1 was challenging. We explored numerous methodologies but generally found the results to be inconclusive. Therefore, we ultimately focused on tuning with the EDML ice core and core U1537 to generate the most robust age model possible. We acknowledge that the information describing the development of the age model may not have been detailed enough, leading to further inquiries by the reviewer. In the revised version, we will include more details to address how each comparative analysis was conducted. For example, the age-depth relationship between Ti and EDML $\delta^{18}O$ records was further refined with comparison from additional proxy records of core PS118_63-1, such as MS, wet bulk density and TOC, which may lead to differing relationships between Ti and EDML $\delta^{18}O$ records as highlighted by the reviewer. Further, we also include information on how each tie point and age uncertainty were selected/rejected.
On the reliability of radiocarbon dating, we recognize the challenges inherent in establishing accurate radiocarbon ages, particularly in this region where uncertainties regarding reservoir age are large. Nevertheless, we are confident in the integrity of our radiocarbon dates, as they are acquired using the MICADAS. This system allows precise gas analysis of carbonates, even when samples are limited in quantity (Mollenhauer et al., 2021).

**RC1.2.1 – Bulk and organic geochemical analyses (section 3.2):** Most of the time the reproducibility of the measures is not mentioned (Opal, TOC, OT-TEX, OT-RIOH). Please detail them in Method section 3.2, whereby the only mention of errors is in the Results for OT.

Author's response: We thank the reviewer for this comment. In the revised manuscript, we now provide information on the analytical precision of the measurements. The methods and essential steps for the determination of the TOC and opal contents as well as any other analytics are adequately documented and cited in the manuscript. Additionally, the methods for determining the $TEX_{86}^{L}$ and RI-OH' indices and their temperature relationships are outlined in the OH-/isoGDGT section.

**RC1.2.2 – Bulk and organic geochemical analyses (section 3.2):** For TEX86L (section 3.2), the authors use the calibration of Kim et al. (2010). This is surprising as they used the calibration of Kim et al (2012) in their recent publications (Lamping et al., 2020; Vorrath et al., 2023).
$SOT_{TEX} = 50.8 * TEXL86 + 36.1$
This must be explained. I wonder how this would alter the TEX86L-based OT record, which is not used in the present study because it does not follow the expected G-IG pattern. The authors claim that TEX86L might be impacted by non-thermal GDGTs, which is fine. However, it must be explained why the non-thermal GDGT sources will only affect the TEX86L (Supplement 5) and not the RI-OH.

Author's response: We thank the reviewer for this comment. However, there may be some misunderstanding in this section. Essentially, we calculated the $TEX_{86}^L$ index according to Kim et al. (2010) as the equation was first introduced in their study. As documented in the original manuscript, the conversion to the OT was done using a modified $TEX_{86}^L$-based OT calibration from Hagemann et al. (2023). Their study extended the core-top datasets used in previous OT calibrations (i.e., Kim et al., 2010; 2012; Lamping et al., 2021) to include more (sub)polar samples from the Southern Ocean, which enabled a more realistic subsurface temperature range as compared to the equation proposed in Kim et al. (2012; -5 to 8°C). We have revised the sentence for more clarity:

*"The isoGDGT-based index, $TEX_{86}^L$ (Eq 2) was calculated following Kim et al. (2010) while the conversion to subsurface ocean temperature (OT; 0 - 200 m water depth; Eq 3) was conducted in accordance to Hagemann et al. (2023)."*

On our $TEX_{86}^L$-based OT record, we like to clarify that this record was not further discussed in the present study primarily due to non-thermal factors, as detailed in the Supplement, and not a result of its deviation from the expected glacial-interglacial pattern. In fact, none of the temperature records from Powell Basin, including diatom-based SSST and RI-OH'-based OT, exhibit any significant glacial-interglacial pattern. Regarding the potential terrestrial impact on RI-OH' paleothermometry, we added a paragraph to address the suitability of RI-OH' as a temperature proxy at our study site.

*"Concerning the RI-OH' approach, the presence of OH-GDGTs has, thus far, only been observed within the cultivated marine thaumarchaeal group I.1a (Pitcher et al., 2011; Liu et al., 2012b; Elling et al., 2014; 2015). Its absence in the terrestrial thaumarchaeal group I.1b (Sinninghe Damsté et al., 2012) suggests a predominantly planktic origin (Lü et al., 2015). While both isoGDGTs and OH-GDGTs are derived from the phylum Thaumarchaeota, variances in their ring composition indicate that the OH-GDGTs may be biosynthesized from different source organisms or differing conditions (Liu et al., 2012b). Additionally, previous studies compared the relationship between various GDGT-based indices (i.e., RI-OH, RI-OH', $TEX_{86}$ and $TEX_{86}^L$) and temperature, and determined that the RI-OH'-temperature relationship shows the most significant correlation in cold-water (<15ºC) regions, making the RI-OH' a robust temperature proxy for the (sub)polar regions (Lü et al., 2015; Lamping et al., 2021; Park et al., 2019; Fietz et al., 2020). Therefore, we suggest that the RI-OH' index holds promise as a potential OT proxy for our study site. However, further work on the distribution of OH-GDGT and calibration studies are still essential to enhance the applicability of RI-OH' as a (paleo)temperature proxy."*

**RC1.2.3 – Bulk and organic geochemical analyses (section 3.2):** The error in the calibration for RIOH is said to be 6°C (Lü et al., 2015). OT variability in cores PS118 and PS67 are within 2°C, which is much lower than the error. How can OT's low variability be robustly interpreted here?

Author's response: We thank the reviewer for this comment and agree that the range of the RI-OH'-based OT temperatures determined for core PS118_63-1 is very narrow. The good agreement between the modern temperature profile (ca. -1°C to 0°C in the upper 200 m water depth) in this area with the reconstructed (youngest/Holocene) temperatures in core PS118_63-1, however, encouraged us to not discard the data. In particular, this is the first application of the novel RI-OH' approach in a marine Antarctic sediment core covering the last glacial-interglacial cycle that allows comparison with numerical model data. Here, we also note that RI-OH'-based OTs – except for the 125 ka BP time-slice – are in overall accordance with the modelled subsurface temperatures. In the manuscript, however, we now draw the reader's attention to the fact that the OT fluctuations are within the error range determined in the global surface sediment calibration study by Lü et al. (2015) and call for attention when interpreting OT variability.

**RC1.3.1 – Diatoms (section 3.3 and 3.4):** I expect that counts are not CRS-free as CRS relative abundances are very variable in core PS118 (0-75%; Fig. 3). It might be good to mention it as regional (Antarctic Peninsula) studies use CRS-free counts to infer environmental conditions. What is the RMSEP for the SSST reconstruction based on IKM336?

Author's response: The reviewer is correct that the full diatom relative abundance is inclusive of the CRS. The RMSEP for the SSST reconstruction based on IKM336 is ±0.83°C, while the standard error of estimate (SEE) is ±0.86°C. We have incorporated the aforementioned points into the revised manuscript:

*"Between 400-600 diatom valves, inclusive of those from Chaetoceros resting spores (Chaetoceros rs), were counted in each sample to ensure statistical significance of the results."*

**RC1.3.2 – Diatoms (section 3.3 and 3.4):** It is not clear to me if the authors used a different transfer function to reconstruct WSIC in core PS67 (IKM172; lines 269-272) while they used MAT336 in core PS118. Sometimes different transfer functions can provide different results and I therefore wonder how robust the comparison between WSIC records in cores PS118 and PS67 is, especially when looking at small changes and timing of rapid changes.

Author's response: Thank you for your comment. We acknowledge the reviewer's concern on the use of different transfer functions (MAT vs. IKM) to reconstruct the WSIC inferred from both cores. We will update the data in the revised manuscript and adopt only the MAT-D274 transfer function for both cores.

**RC1.4.1 – Numerical model (section 3.5; Supp S3):** The authors use here a single model (COSMOS) with, as far as I understood, PMIP3 paleogeography for glacial settings. I wonder whether (1) one model is sufficient; (2) how COSMOS performs compared to other PMIP models, especially in terms of sea-ice dynamics which is the weakness of most ESMs; (3) why not using PMIP4 settings; (4) which ice-sheet configuration was used (GLAC-1D (Argus et al., 2014, Peltier et al., 2015; ICE-6G_C, Ivanovic et al., 2016); (5) is it sensible to use a similar paleogeography at PGM and LGM, and at LIG and Holocene (Table S1)?

Author's response: We thank the reviewer for his inquiries regarding the employed climate model simulations. We take these inquiries as a guide to improve the revised manuscript on the presentation of model setup and on the discussion of the results. Please find below our point-by-point reply to the respective remarks by the reviewer:

(1) and (2): The reviewer provides a valid argument regarding the use of one single model in our study. It is indeed a well-known fact that a multi-model ensemble often shows better skill than individual model components. This is due to the fact that in an ensemble of models, various biases in individual models cancel out each other to some degree (e.g., Shi et al., 2023). Our justification to the use of COSMOS as the only model in our study has been a pragmatic one – aiming at analyzing model outcomes at specific glacial and interglacial time slices: PI, 6, 21, 125, 128 and 140 ka BP. While PMIP3 and PMIP4 provide results to some of the time slices studied by us (namely PI, 6 and 21 ka BP), their coverage of the last interglacial period primarily focuses on 127 ka BP, which does not align with our target time slices of 125 and 128 ka BP, and a coordinated model setup for the penultimate glacial period is absent. Consequently, COSMOS must remain the pivot of the modelling component of our study. Fortunately, COSMOS has been extremely well-tested across a vast range of climate states, including current climate and both much colder and much warmer climates than today. A study by Lunt et al. (2013) has shown that COSMOS is, in an ensemble of PMIP3-grade models, among those models with a comparably minor bias of sea surface temperatures in the Southern Ocean (cf. their Fig. 4 e,f). This inference somehow recommends use of this model if we are focusing on sea ice cover and temperatures in the Southern Ocean. We acknowledge that documenting these points is something that can be improved in our manuscript.

To this end the revised version of the manuscript will contain a review of the use of COSMOS in previous work and of the model's performance in comparison to other models. Furthermore, we provide, as an additional "data item", information related to an additional dedicated comparison of results of our study to those available from PMIP3 and PMIP4. However, we note that due to limited coverage of PMIP3 and PMIP4 regarding relevant time slices, the comparisons made will be focused on PI, 6, 21 ka BP as well as 127 ka BP against that of 125 and 128 ka BP. To this end, we note that our model performs very well against the PMIP3 simulations. Although there are some disagreements with PMIP4 results, we find that these disagreements are not due to a weakness of our model. These findings will be evaluated and discussed in the revised manuscript.

(3) Indeed, COSMOS simulations presented by us largely stem from a PMIP3 framework. This is a side effect of the long use of the model which, and we would like to stress this point, is the one and only reason why we are actually able to cover the whole study period with results from model simulations. Neither PMIP3 nor PMIP4, and also none of our own work done with PMIP4-related models, can compete regarding coverage of all relevant time periods in this study. We provide a discussion of this topic in the revised manuscript. Our model shows some differences with respect to PMIP4 simulations

where a meaningful comparison is possible (this excludes the time slices of the Last Interglacial and of the Penultimate Glacial Maximum). Yet, based on previous literature and on the nature of the disagreement between our model and the CMIP6 ensemble, we infer that our modelling provides robust results in the context of this study. This argumentation is provided in the revised manuscript.

(4) We apologize for not properly documenting details of the ice sheet configuration. Indeed, the model setup is based on the PMIP3 LGM blend of various ice sheet reconstructions as outlined by Abe-Ouchi et al. (2015). We have amended the revised manuscript to convey this information:

*"For interglacial climates we employ a modern geography. The boundary conditions for the Last and Penultimate Glacial Maximum have been set up for a study by Zhang et al. (2013) based on the PMIP3 modelling protocol. Details of the ice sheet reconstruction, that is a blend of ICE-6G v2.0 (Argus and Peltier, 2010), ANU (Lambeck et al., 2010) and GLAC-1a (Tarasov et al., 2012), are described by Abe-Ouchi et al. (2015)."*

(5) Indeed, the use of same geographies for mid-Holocene and Last Interglacial is an approximation. The same holds for Last Glacial Maximum (LGM) and Penultimate Glacial Maximum (PGM) ice sheets. Otto-Bliesner et al. (2017) note that, while there are indications that the Last Interglacial ice sheets showed characteristics that differ, there is still ample uncertainty regarding how exactly a Last Interglacial ice sheet was configured. As a consequence, the PMIP4 modelling protocol for Interglacials (Otto-Bliesner et al., 2017) refrains from modifying any ice sheets for the Last Interglacial time slice simulation at 127 ka BP from their present configuration. Our decision to use the same ice sheet configuration for mid-Holocene and Last Interglacial is therefore consistent with the PMIP4 protocol. For our decision to use the same ice sheet configuration for LGM and PGM a similar statement holds: while there is indication that ice sheets may have differed between both time slices (e.g., Rohling et al., 2017, who present in their Table 1 a summary of published (and diverging) sea level contributions of the Antarctic Ice Sheet for the PGM), details of differences in ice sheet configuration are unclear. Therefore, there is ambiguity on how exactly the difference in ice sheets should be configured as model boundary condition.

**RC1.4.2 – Numerical model (section 3.5; Supp S3):** As PS118 core is close to Antarctica I expect any change in ice-sheet paleogeography can strongly alter the results. More specifically, it is mentioned that WAIS may have partly collapsed during the early LIG, which never occurred during the Holocene. I wonder (1) what the impact of this partial collapse on Powell Basin – Scotia Sea oceanography, sea ice, and productivity, and (2) if it could reconcile simulated and data temperature (SST and OT, cf comment on the Discussion).

Author's response: We thank the reviewer for the additional inquiries. In the revised manuscript, we document the motivation for our choice of ice sheet boundary conditions based on previous work. The reviewer has rightly noted that differences in ice sheet configuration may impact on results of sea ice state and oceanography in the Southern Ocean – a detail that is unfortunately neither represented in relevant simulations from PMIP models nor in our modelling work. We therefore highlight in the revised manuscript that sensitivity studies with plausible alternative ice sheet configurations are an important focus for future work.

**RC1.5.1 – Discussion (section 5.1):** I found the overall reconstruction, interpretation, and argumentation valid. My main concern is about the fact that the sub-surface temperatures (based on OT-RIOH) are lower than sea-surface temperatures (based on diatom transfer function) in core PS118 (SST > OT), which is not supported by modern ocean conditions (please show vertical temperature profiles from both core sites) or the model (SST < OT). Given the 6°C error on the RIOH calibration, I wonder whether it is sensible to interpret absolute OT data, and if the 1°C difference between OT and SST is true and significant. For example, OT is as low at ~125 kyr BP as during the glacial periods (Fig 7, right column), which is not substantiated by the model (Fig 7, left columns) or by any of the other proxies presented (PIPSO, SST, WSIC, Productivity, SAT) in any of the cores (PS118, PS67, ice cores).

Author's response: We agree with the reviewer that the GDGT-derived OTs are lower than sea surface temperature (diatom-based SSST). We attribute this to the different water depth ranges each proxy is calibrated to (e.g., GDGT-derived OT represents temperature at water depths between 0 and 200 m, whereas diatom-based SSST reflects surface temperature between water depths 0 and 10 m). We revise the statement regarding a comparison between the OT and SSST, and include a vertical

temperature profile from the Powell Basin, close to the site of core PS118_63-1, that validates our (SSST > OT) results.

Concerning the use of absolute OT, we acknowledge the complexities involved in interpreting our OT results based solely on absolute values, particularly given the wide (±6°C) residual temperature range. However, as we are comparing it with other temperature data, such as diatom-based SSST and model-simulated (sub)surface temperatures, we maintain the use of absolute OT for comparability and consistency across our analyses. Moreover, our study primarily focuses on interpreting environmental variability over glacial-interglacial timescales. Therefore, whether OT is expressed as absolute values or anomalies does not significantly impact our interpretation as the underlying variability trend inferred from the proxy record remains the same. Nevertheless, we will review our original manuscript and pivot our discussion of OT towards evaluating variability trends rather than on the absolute values.

**RC1.5.2 – Discussion (section 5.1):** There is a long discussion associated with the difference between OT and SST, calling on the bipolar seesaw, recirculation of WDW, or melting of the ice sheet (lines 565-575; 630-641 for example) that are not substantiated by the data for the reasons expressed above. Especially, when looking at short-lived events in a chronological framework with several thousands of years of imprecision. The link to NH processes might completely change if records are moved by a couple of thousands of years. Overall, these interpretations are at odds with the model output, which I reckon may have some flaws too, but I would recommend simplifying the discussion on this specific point. Overall, the Discussion is lengthy and tedious. I would suggest to simplify it.

Author's response: We thank the reviewer for your recommendation. The discussion will be revised to incorporate the suggestions.

**RC1.6 Additional minor comments: a – l**

a) The rule is to write "sea ice" when a noun (reconstruction of sea ice) and "sea-ice" when an adjective (sea-ice concentration). Authors use alternatively "sea-ice concentration" and "sea ice concentration". Please harmonise throughout the manuscript.

b) Lines 237-238 and elsewhere: Please change F. cugr with either *F. curta* gp or FCC as in previous publications.

c) **\*Line 313 and elsewhere**: I would refrain from using %TOC and %BSi as a direct measure of productivity in contouritic systems near islands. Secondary processes such as dilution, transport, and dissolution are extremely important in this setting. I wonder whether fluxes would not be a better metric.

d) Line 347 and elsewhere: *R. leventerae* (not R. leventarae). Additionally, use italics for diatom species.

e) **\*Lines 480-481**: I do not understand what you mean. Is it that the seasonal variations are in the same range as G-IG variations? And what? I am not sure you use this afterward in the Discussion.

f) Line 487: The phrasing is a bit optimistic. These values are not calculated but "attributed" based on almost barren diatom samples for WSIC and the absence of Diene and Triene for PIPSO. So it is not a real quantification. It is mentioned in the Results, but it must be mentioned again here in the Discussion.

g) Line 527: Replace caverns with cavities.

h) **\*Lines 544-545**: I do not understand what you mean. That the large decrease in sea ice in the Atlantic sector, modeled by Holloway et al. (2017), is a summer signal?

i) Lines 569-570: There is no mention of the Weddell Sea in Marino et al., (2015). Over-interpretation of this study.

j) **\*Lines 720-723**: I do not see the saw-toothed pattern in BSi and TOC across TI (Fig. 3), but I agree that the magnitude of changes appears more important at MIS6-5 than at MIS2-1.

k) **\*Lines 734-737**: This statement is true from the PIPSO, BSi, and TOC point of view in core PS118. There are however no significant differences in WSIC between the LIG and the Holocene in this core (Fig. 4). Can we infer that sea-ice seasonality was greater at the LIG? Can it be linked to different seasonal distributions of the regional insolation (Bova et al., 2021)? A greater sea-ice seasonality is however not corroborated by the simulated sea-ice maps (Fig. 5).

l) **\*I also note that** BSi changes appear larger across TI than across TII in core PS67, maybe also changes in SSST (Fig. 4). How does it fit with the interpretation based on PS118 data?

Author's response: We thank the reviewer for the meticulous feedback and will consider all recommendations in the revised manuscript. Additional responses to specific comments (\* in bold) are also enclosed below:

\* c): We acknowledge that expressing biogenic opal and TOC contents as flux rates can be advantageous in contouritic systems near islands. However, due to the lack of robust age constraints for the upper part of sediment core PS118_63-1, we refrain from the determination of accumulation rates. Given that our study focuses on (paleo)environmental variability from a glacial-interglacial perspective, we believe that – in line with other publications – using % content for TOC and opal, in our study, is sufficient to describe the region's glacial-interglacial productivity variability.

\* e): We want to highlight that while there is a G-IG temperature difference at both core sites, no seasonal variation is observed at each core site across each time slice. We amended the statement to provide more clarity:

*"As for the simulated OT, the model displays a ~1.6 and ~3°C glacial-interglacial variation at core sites PS118_63-1 and PS67/219-1, respectively, but no appreciable OT change is observed between the winter and summer seasons, respectively, of each time slice (Fig. 7)."*

\* h): As also the WSI record from marine core PS2305-6, located slightly north of our core site, indicates the presence of WSI during MIS 5e (Bianchi and Gersonde, 2002; Gersonde and Zielinski, 2000), we re-phrased this section for clarity:

*"We assume that the modelled winter sea-ice retreat seems to be valid for more distal ocean areas, whereas at the core sites in the Powell Basin and the South Scotia Sea, ice-sheet derived meltwater may have acted as a driving mechanism fostering local sea-ice formation during winter, which is not captured by the simulation in Holloway et al. (2017)."*

\* j): We intended to describe the gradual increase with periodic fluctuations observed in both proxies across TI. The statement is rephrased for better clarity:

*"In contrast, TI exhibits a pattern characterized by a gradual increase with periodic fluctuations throughout the termination for both TOC and biogenic opal content."*

\* k): From the records, we could infer that the sea-ice seasonality is greater throughout the LIG. However, the mean annual insolation for the region for the LIG is slightly lower than for the Holocene (Laskar et al. 2004), hence, we cannot justify that the warmer and less ice-covered LIG is a result of the (seasonal) insolation (Bova et al, 2021). We added the following statement to the revised manuscript:

*"However, no significant difference in the WSIC between both interglacials was noted. The discrepancy in warming intensity likely occurred seasonally and coincided with maximum summer insolation (see also Fig. 4 in Bova et al., 2021). Nonetheless, a lower mean annual regional insolation (-1.1 W/m$^2$ difference; Laskar et al., 2004) during the LIG does not explain the warmer and less ice-covered conditions reconstructed for the region."* "

Thank you for pointing out the similarity in the modelled sea-ice distribution for the 125 ka BP and 6 ka BP time-slice. The reviewer is correct in assuming that the same model boundary conditions have been applied for both interglacial time slices. At present, the significant uncertainties stemming from proxy reconstructions for the LIG present challenges for models in accurately determining the boundary configuration for simulating past sea ice in the Southern Ocean. In the revised manuscript, we will provide more information pertaining to our motivation on the model boundary condition used and improvements for future work. We also refer to the responses provided for RC1.4.1 (5) and RC1.4.2.

\* l): While the increase in biogenic opal content in core PS67/219 is faster during TII than during TI, it's less pronounced in terms of absolute concentrations (i.e., 15% to a peak value of 41% during TII;

12% to a peak value of 40% during TI). For SSSTs in this core it's similar: an abrupt though smaller rise in temperature during TII compared to TI. However, the average BSi and SSST across both termination intervals indicate higher values for TII (35%; 0.7ºC) compared to TI (26%; 0.5ºC). Unfortunately, the BSi and SSST records in core PS118_63-1 are inconclusive, with a mean BSi content of 6% for both terminations, and an incomplete SSST record due to poor preservation during the early stages of both terminations. Nonetheless, alternative proxies for productivity (TOC) and ocean temperature (OT) reveal a similar trend in the Powell Basin: TII (0.5%; 0ºC) was warmer than TI (0.4%; -0.3ºC). The delayed increase in biogenic opal during TII and also TI in core PS118_63 compared to PS67/219 may be attributed to the latter being located further to the north - so this site would experience a warming/sea ice decrease and hereby triggered increased biogenic opal production a bit earlier than site PS118.

**RC1.7 Figures: a - c**

a) Figures 3-4: Please label the plots in each figure (Fig. 3A, etc…) and refer to full labels in the main text to ease reading.

b) Figures 5-7: Please reverse plots with PS118 data below PS67 data to fit the latitudinal distribution of the core. It will probably be easier to follow.

c) Figures S4-5: Please label each record.

Author's response: We thank the reviewer for the recommendations and will update the figures accordingly in the revised manuscript.

---

## Referee Report (RR1)

I believe that the authors provided adequate answers to most of my comments and I am generally happy with the corrections. I only have minor comments related to:

The vertical profiles of ocean temperature (Figure 1): Which season do they represent? I believe it is important information as the temperature profile may change through the year, with the development of a strong thermocline in sub-surface (25-50 m) in summer resulting in lower temperature at depth than at the surface (Foldvik et al., 1985; Martinson et al., 2008; Venables et al., 2013; Vorrath et al., 2023). I however reckon that it may depend on the region (Muench et al. 1990) and I do not know how the vertical structure in the Powell Basin evolves throughout the year. I would recommend presenting a spring-summer vertical profile as the diatom transfer function provides spring-summer SST while the RI-OH is understood as summer OT (Vorrath et al., 2023). In this vein, I do not understand why the same team refers to the Ri-OH OT as a summer signal in Vorrath 2023 and here as an annual signal. This discrepancy must be explained and, if OT is lower than SST in summer, the discussion about the vertical stratification probably needs to be re-evaluated.

Figures: I would recommend adding the simulated values (SSI, SWI, summer SST, summer OT) at both core sites on the temporal plots (right column of each figure) for each time slice. This will help grasping the similarities or discrepancies between the data and model output. For example, the color increment is 1°C in Figure 6, which does not allow a quick comparison.

[Figure]

I also realized that I did not understand the last sentence of the caption in figures 5-7. The proxy-derived data are shown by the curves, not by the shaded areas as it written (or at least as I read it).

---

## Author Response (AR2)

**Author's response to second round of review**

Dear Editor and Reviewer,

We thank you for your continued guidance and feedback to improve our paper. We carefully considered the comments and now provide a marked-up manuscript detailing the revisions and corrections. A "clean" and reader-friendly version of the revised manuscript and supplementary materials are also uploaded. Please see below for our detailed response (in red) to the reviewer's comments.

**Reviewer's comments:**

I believe that the authors provided adequate answers to most of my comments and I am generally happy with the corrections. I only have minor comments related to:

The vertical profiles of ocean temperature (Figure 1): Which season do they represent? I believe it is important information as the temperature profile may change through the year, with the development of a strong thermocline in sub-surface (25-50 m) in summer resulting in lower temperature at depth than at the surface (Foldvik et al., 1985; Martinson et al., 2008; Venables et al., 2013; Vorrath et al., 2023). I however reckon that it may depend on the region (Muench et al. 1990) and I do not know how the vertical structure in the Powell Basin evolves throughout the year. I would recommend presenting a spring-summer vertical profile as the diatom transfer function provides spring-summer SST while the RI-OH is understood as summer OT (Vorrath et al., 2023). In this view, I do not understand why the same team refers to the RI-OH OT as a summer signal in Vorrath 2023 and here as an annual signal. This discrepancy must be explained and, if OT is lower than SST in summer, the discussion about the vertical stratification probably needs to be re-evaluated.

Author's response: We thank the reviewer for the helpful suggestion. The original temperature profile we used represented the annual profile for both core sites. After careful consideration, we have adopted the reviewer's proposed spring/summer vertical profile. This adjustment offers a surface temperature profile that aligns more closely with the diatom-based proxy records for both core sites. In the Scotia Sea (PS67/219-1), the spring/summer temperature remains warmer than the annual mean temperature down to a water depth of ca. 50 m. Below this depth, the spring/summer temperature and the annual mean temperature exhibit a congruent profile down to 2000 m water depth (see Fig. 1 below). In the Powell Basin (PS118_63-1), the spring/summer temperature remains warmer than the annual mean temperature even down to 150 meters water depth, which may point to a deeper mixing during summer. Below this depth, both temperature profiles are identical down to 2000 m water depth.

[Figure]

*Figure 1. Vertical temperature profiles (annual vs spring/summer) for core sites PS118 and PS67.*

The RI-OH'-derived ocean temperatures for core PS118_63-1 (ca. -2 to 0 °C) are in the range of both the modern spring/summer and annual mean temperatures. The reviewer's assertion that RI-OH' temperatures signify a summer signal is contradicted by a comparison of summer insolation values with the OT record (Fig. 2, below). In the case of Vorrath et al. (2023), the authors observed a similar trend between their RI-OH' OT, diatom-derived SSSTs and summer insolation (see Fig. 5 in Vorrath et al., 2023) and thus proposed that the RI-OH' OT could reflect a summer signal. However, at our core site, we do not observe this correlation between RI-OH' OT and summer insolation (see Fig. 2 below). Instead, we propose that the observed OT variations may be influenced by changes in ocean circulation as well as dense and cold shelf (and melt) water advection between 0-200 m water depths. Also, as the RI-OH'-SST calibration equation was derived from the World Ocean Atlas mean annual SST datasets (Fietz et al., 2013; Huguet et al., 2013; Lü et al., 2015), we here prefer to report the RI-OH' OT as reflecting an annual signal and note that further efforts in developing seasonal and/or regional calibrations for hydroxylated (and also iso) GDGTs are needed to overcome this uncertainty.

[Figure]

*Figure 2. RI-OH'-derived temp at core site PS118_63-1 vs mean summer insolation.*

Figures: I would recommend adding the simulated values (SSI, SWI, summer SST, summer OT) at both core sites on the temporal plots (right column of each figure) for each time slice. This will help grasping the similarities or discrepancies between the data and model output. For example, the color increment is 1°C in Figure 6, which does not allow a quick comparison. I also realized that I did

not understand the last sentence of the caption in figures 5-7. The proxy-derived data are shown by the curves, not by the shaded areas as it written (or at least as I read it).

Author's response: We thank the reviewer for the suggestion. We have incorporated the suggestions to improve the figures and revised the respective caption.